# Exploring climate smart agriculture in Turkey: Enhancing food security and sustainable practices for the reduction of CO$_2$ emissions

**Nawaz Khan** [ID]*, **Wang Jie**

 School of Internet Economics and Business, Fujian University of Technology, Fuzhou, Fujian, China

* 61202008@fjut.edu.cn

## Abstract

Climate-smart agriculture entails the reduction of CO$_2$ emissions, adaptation and modification of technology to enhance resilience to climate change, and sustainable increase of incomes. This study evaluates the effectiveness of smart agricultural practices in Turkey, which significantly impact food security and mitigate CO$_2$ emissions. The decoupling technique was implemented to estimate the portfolio returns and examine their correlation with climate-smart agriculture and CO$_2$ emissions over the anticipated period of 1992–2023. The decoupling technique was implemented to accomplish the two primary objectives. First, it is employed to calculate the percentage change in portfolio returns that is linked to both high and low weighted risk allocations. Second, it enables the prediction of CO$_2$ emissions levels for the next five years, which are influenced by sustainability practices and food security fluctuations. A corresponding difference in the efficacy of climate-smart agriculture has been demonstrated in the agricultural context by a percentage change in continuous, single aeration, and multiple aeration practices. The estimated results suggest that the decoupling trend in portfolio returns is significantly influenced by factors such as rice cultivation, field rise, and soil management, which also contribute to the highest weighted risk. Additionally, this factor consistently shows the highest weighted importance in determining overall portfolio returns, as it exhibits the largest marginal effects. Consequently, this investigation substantiates the detrimental influence of these variables on CO$_2$ emissions. Turkey's sustainable smart agriculture process is essential for the efficient expansion of the economy, as it integrates climate change considerations into current policies and initiatives and reinforces the policy indicator of economic consideration with environmental protection in terms of CO$_2$ emissions.

## 1. Introduction

Climate-smart agriculture (CLS-A) reduces CO$_2$ emissions by lowering the emission of CO$_2$ from agricultural activities and helping to sequester CO$_2$ from the

**Data availability statement:** All relevant data are within the paper and its Supporting Information files.

**Funding:** The author(s) received no specific funding for this work.

**Competing interests:** The authors have declared that no competing interests exist.

atmosphere [1]. Cover cropping practices sequester $CO_2$ and contribute to year-end soil organic matter, while enhanced grazing management increases carbon storage in climate-smart agriculture (CLS-A) [2]. First, Turkey is very susceptible initially to environmental changes [3]. It is in the Mediterranean basin, where increases in temperature as well as droughts due to extreme weather are already happening [4,5]. Furthermore, CLS-A helps to shape the economy and uses sustainability in agriculture. As such, CLS-A practices resilience and buffers environmental change, making food security and a stable economy possible in the future. Secondly, we must acknowledge the significant impact of greenhouse gas (GHG) emissions, particularly $CO_2$ emissions [6]. By implementing strategic CLS-A measures that reduce emissions and support soil management and agroforestry, Turkey can significantly contribute to its efforts to meet and transition to a sustainable economy. Furthermore, GHG emissions in 2021 were 564.4 million tonnes (Mt), which was 7.7% higher than the previous year. The total GHG emissions per capita were estimated to be 4 tonnes of $CO_2$ in 1990, 6.3 tonnes of $CO_2$ in 2020, and more than 0.4 tonnes in 2021 [7]. The agricultural sector emissions were estimated to be 72.1 Mt $CO_2$ eq. in 2021, representing a 56.5% increase from 1990. However, the emissions from the previous year fluctuated by 1.5%. In this instance, there has been a decoupling of CLS-A. We have decoupled the development and rapid production processes that are harmful to the environment. Correspondingly, it provides a framework for Turkey to modernize and reinforce its CLS-A sector and thereby benefit the environment and economy [8,9].

The CLS-A is a holistic approach designed to transform agricultural systems to effectively and sustainably support food security and development in a changing climate. It integrates three main pillars: increased productivity, enhanced resilience, and reduced GHG emissions. To address the interconnected challenges of resources, food security, and climate change in Turkey, CLS-A must adopt a comprehensive approach designed to transform sustainable agriculture sectors and increase resilience. In Turkey, the justification for CLS-A is based on three primary integrated objectives: productivity, adaptation, and mitigation. The Turkish agriculture sector is currently grappling with the tangible consequences of climate change and revolution [10]. Furthermore, this agricultural sector is in dire need of sustainable competitiveness and productivity to satisfy domestic and global food demand, as well as to increase Turkey's agricultural food export opportunities in the international market while reducing GHG emissions. Turkey has implemented the CLS-A, which includes advanced technology, farmer education, and policy support as necessary components [11]. These implementations can be classified into essential practices, including institutional support and financing, land and soil management, water management technologies, and technological modifications in farming [12]. Additionally, Turkey, a region highly vulnerable to climate change impacts, is increasingly focusing on CLS-A to address its agricultural challenges, ensure food security, and contribute to global climate goals. Turkey's agricultural sector is a major source of jobs and money, but it has some significant problems with soil degradation, food security, and GHG emissions. The degradation of soil poses a threat to the country's long-term sustainability [13]. Agricultural food security has significantly increased reliance on imports and has

contributed to higher food prices. In 2023, the food inflation rates were over 55%, and the agriculture sector contributed 71.5 Mt $CO_2$ equivalent in 2023, which is an increase of 37.9% from 1990 [14]. We examine the environmental protection implications of CLS-A through agriculture, which involves the integration of trees into agricultural systems to achieve two objectives: the limitation of forest denudation and the sequestration of atmospheric $CO_2$ in biomass and soil [15,16]. The CLS-A also approaches productivity and income regarding agriculture, in which sustainable intensification and innovation systems play the most important role in the climate. Climate-smart technology and practice control environmental degradation in the public and private sectors and help farmers facilitate technology transfer and knowledge sharing [17]. Second, enhanced resilience shows that conservation agriculture and diversification of farming systems regarding crops and livestock reduce the risk associated with climate impacts [18]. Third, mitigation involves reducing emissions through livestock, soil, and land use management. Lastly, Turkey can move forward with the technological revolution in agriculture, and also the policy development and innovative financing mechanisms will be crucial for the widespread adoption of CLS-A [19].

This study addressed CLS-A and decoupling in accordance with the two sequences. First, this study addresses the global context of CLS-A and decoupling with a triple-win approach, which includes productivity, adaptation, and mitigation. CLS-A functions as a global framework for sustainable agriculture, continuously flooded through single and multiple aerations. Nevertheless, decoupling is a mandatory analytical objective that is essential for separating the agricultural output of individual indicators from environmental pressures such as GHG ($CO_2$ emissions). Additionally, CLS-A strategies are calculated quantitatively through the decoupling process in terms of average return and total risk. Second, Turkey is extremely susceptible to environmental change owing to its agricultural sector, which includes the conflict between climate and food security. An ideal CLS-A can be achieved by concurrently reducing the carbon footprint of agriculture and implementing measures for environmental resilience.

The CLS-A practice has three primary objectives: to minimize climate change and then maintain the sustainability of this aim [20,21]. The second is sustainable agriculture, which focuses on production methods that reduce $CO_2$ emissions. In response to climate challenges, this objective is to increase productivity and income, as well as to ensure food security and improve agricultural yields [21–23]. Additionally, transitioning agriculture to effectively support development and reduce GHG emissions, which in turn reduces the contribution of agriculture to climate change by reducing its carbon footprint. The third primary objective of this study is to provide a comprehensive analysis of the decoupling technique, which is used to determine the estimated portfolio returns method and its relationship with CLS-A. Decoupling is viewed as a potential solution for improving food security and reducing $CO_2$ emissions in Turkey. Additionally, the CLS-A project shows that decoupled development has led to a rise in environmental costs, like GHG emissions. Consequently, sustainable agriculture products are included in this portfolio through a decoupling process. Lastly, these objective contributions demonstrated, by means of the application of decoupling techniques, the important contribution to $CO_2$ emissions as well as the vulnerability to climate change [24,25]. This study argues that by decoupling, Turkey can meet its food security goals with limited environmental impacts [26]. To investigate and implement effective decoupling techniques, we need a multi-pronged approach that considers such technological developments [27]. It may be possible to decouple these goals: by increasing food production, we can stop our environmental impact [28]. There are several ways to achieve this. Second, agricultural practices such as enhanced agricultural practices, including integrated pest management, conservation tillage, and precision CLS-A, have been tested to increase the crop yield while reducing water consumption, soil erosion, and dependence on pesticides [29]. Second, eco-friendly innovations: water for irrigation can be sourced from salt water, drought-resistant crops can be grown, and a vertical farming approach can be adopted, which will technically work around the constraints of available resources and lead to more efficient food production.

Third, supply chain issues lead to significant food waste. Reducing food waste from farm to fork is one approach that can help with decoupling. Lastly, a significant portion of the food waste occurs throughout the supply chain. The results of this study suggest that the decoupling (portfolio returns) of the agricultural sector in Turkey could have a substantial

influence on the reduction of $CO_2$ emissions and the improvement of food security. Decoupling can foster multifarious, sustainable, and fruitful agricultural methods by empowering producers to make decisions more in line with market forces. Turkey would benefit from implementing supplementary policies that encourage producers to cultivate crops using sustainable methods, thereby slowing the process. Transitioning to a decoupled technique could be a substantial enhancement for Turkey in addressing the challenges of food security and high levels of $CO_2$ emissions [30]. This CLS-A research aims to improve low-cost and long-term farming systems. It could involve finding new sustainable food buyers or verifying fair trade [31]. As a result of decoupling, rural infrastructure, healthcare, and education are more readily accessible. The implementation of waste reduction techniques has achieved significant progress toward decoupling. By implementing these measures, Turkey can guarantee its food supply in the future and improve the global environment [32,33].

This study emphasizes CLS-A's climate change adaptation through research. This underlines the need to focus on relevant research areas such as drought management, climate change effects, adaptation, and evidence-based food security. This also emphasizes the need to collect and verify climate change data for each river basin and region. Authorities can affect future event preparations after considering the data. Research and development aim to improve land, water, and crop management in response to climate change. Therefore, the literature highlights the specific ideas and actions required to achieve this goal. This study emphasizes the need to examine how climate change may affect CLS-A output, production, and local information. The main goal is to monitor climate change and create measures to limit its effects on agriculture in the region. This study examines whether sustainable farming can reduce the impact of climate change on Turkish food production. We investigated Turkey's environmental and economic policies using critical economic facts, global choices, and laws. According to studies, Turkish farmers can slow global warming by adopting modern farming methods. The legal application of Turkey's program reduces hazardous emissions and boosts economic growth and environmental security. Economic data can illustrate the financial feasibility of ideas.

This study comprises two sections: background information and research on the environmental implications of decoupling. Section 3 describes the research goals, theoretical and conceptual frameworks, methodology, and results. The estimation and results are presented in Section 4. Section 5 presents the results. The conclusion summarizes the findings of the study and suggests relevant policies.

## 2. Literature review

### 2.1. Climate-smart agriculture

A number of sustainable CLS-A-related policies and projects have been established by the Turkish government [34,35]. However, additional efforts are required to ensure the effectiveness of these policies and initiatives. To create new sustainable agricultural technologies, the agricultural sector must engage in research and development. How Turkey deals with its $CO_2$ emissions problems will determine the fate of its CLS-A economy [36]. In addition, the Turkish Ministry of Agriculture and Forestry (TMAF) has stated that its objective is to enhance the sustainability of agricultural development and the competitiveness of agricultural food security. The CLS-A has three most critical components, which provide a comprehensive global perspective: productivity with food security and income, the adoption and development of resilience to environmental change, and the reduction of $CO_2$ emissions, as clearly identified by a framework [37]. As long as the solutions are within reasonable costs, individuals who are responsible for molding policies and farmers can identify the methods that effectively capture the most $CO_2$ emissions [38]. It is imperative to investigate the factors that influence individuals' decisions to adopt ecological and dining behaviors. [39]. More sustainable methods can help Turkey's CLS-A endure the impacts of climate change, water scarcity, and soil degradation [40]. Consequently, this sector will persist in the future, guaranteeing that the Turkish populace has an adequate supply of sustenance [41]. Sustainable growth also supports the transition to transformative change in smart agriculture through various initiatives, such as innovation, capacity development, and technological revolution in the context of smart farming. Currently, there is insufficient information available to determine whether various decoupling strategies are the most effective in mitigating the impact of climate change on

CLS-A. Furthermore, it is imperative to conduct additional research to determine the most cost-effective decoupling solutions [42]. An integrated method that transforms agriculture and mitigates environmental change, the CLS-A demonstrated climate-resilient practices and reduced GHG emissions in the previous study [43]. Therefore, developing a strategic plan for agricultural balancing is necessary. Additionally, CLS-A adoption in Turkey is encouraged, and the potential for sustainable agricultural growth is increased [44,45]. However, we examined CLS-A in this study in relation to food security and sustainable practices for lowering $CO_2$ emissions. Research is necessary to develop effective laws that promote environmental protection, prevent detrimental actions, and ensure equal opportunity in agriculture [46,47]. Furthermore, research is essential to explore the effects of current and future policies.

## 2.2. Sustainable practice of Turkey and $CO_2$ Emissions

Turkey can address these challenges and fill these knowledge gaps by implementing a successful decoupling strategy that is financially feasible, socially acceptable, and sustainable. It guarantees the availability of food for individuals and the sustainability of food systems in the future. Despite the adoption of CLS-A, certain research areas require improvement to achieve more favorable outcomes and acceptance. Currently, there are numerous critical regions in the globe [48]. We must determine whether CLS-A is economically feasible for small and medium-sized farms in Turkey [49]. The CLS-A is a global strategy to increase food productivity in the face of climate change and lower GHG emissions, thereby making it more climate-resilient. It is crucial to sustainable agriculture, and because developing nations are most negatively impacted by environmental change, they lack the adaptive capacity to deal with its repercussions [50]. To facilitate the successful implementation of sustainable practices that capitalize on emerging technologies, farmers must receive assistance in improving their digital literacy [51]. Research-informed subsidies and tax breaks are two effective policies that motivate farmers to adopt sustainable practices [52]. To incentivize producers to embrace market mechanisms, sustainable practices such as certification systems or premium pricing for sustainably produced food should be investigated. Filling these knowledge shortfalls in agricultural sustainability can hasten Turkey's shift to ecologically friendly farming practices [52].

## 3. Methodology

This study implemented CLS-A and decoupling, which are strategically interconnected in the global endeavor to accomplish sustainable development and combat climate change in Turkey's agricultural sector. One of the primary objectives of decoupling is to disrupt the positive correlation between variables such as field burning, managed soil, urea application, manure management, enteric fermentation and rice production. This is achieved through the implementation of decoupling techniques and their subsequent environmental repercussions. In addition, the total and average risks were individually calculated for continuously flooded, single, and multiple aeration systems. This context pertains to Turkey's agricultural sector and environmental change, including GHG emissions, while simultaneously diminishing the association and connection between environmental impact and impact. Under the new climate change policies and realities, the CLS-A approach aims to reorient and transform the agricultural sector to support sustainable development and food security. Additionally, this method provides an informative tool for determining environmental policies, agriculture trade, resilience, and allocation. Furthermore, it is consistent with sustainable development and market conditions, which has facilitated the development of a portfolio decoupling technique designed to reconcile environmental resilience with financial performance.

### 3.1. Decoupling and portfolio technique

This segment employs a dynamic development model of Turkey's global economy to examine the interplay between food security and the environment. In this section, we analyze decoupling as a potential solution to improve food security and reduce $CO_2$ emissions in Turkey. In the context of agriculture and $CO_2$ emissions as environmental factors, the

fundamental approach of this study is to decouple the determinants of portfolio returns. The decoupling technique was employed to compile the data in this study, and subsequently, individual portfolio returns were calculated. This illustrates the correlation between rice production, urea application, soil management, field burning, enteric fermentation, and manure management in Turkey and GHG emissions. $CO_2$ emissions were individually analyzed in terms of GHG emissions. However, policymakers must confront the uncomfortable reality that Turkey's agricultural sector is economically untenable. The primary objective of this investigation is to evaluate the return of each portfolio following the decoupling process, which is conducted in accordance with the three steps specified. First, the portfolio technique considers environmental factors and agricultural shocks as random market risks, thereby isolating their impact. We directly attribute the climate events—droughts, food, consumer preferences, agriculture policies, and sustainable agriculture products in this portfolio—through the decoupling procedure. Consequently, this method is more detailed in its ability to comprehend the fundamental portfolio change in the context of high- and low-weighted risks in numerous, continuously inundated, and single aeration scenarios. Secondly, this technique also enables us to gain a more comprehensive understanding of the environmental degradation caused by $CO_2$ emissions in the agricultural system. Specifically, we can directly discern how commodity market prices influence the performance of sustainable farming practices. Thirdly, we examine the investment strategies within the agricultural sector. Develop more targeted risk mitigation strategies for environmental and agricultural vulnerabilities through portfolio returns. In this regard, it is straightforward to distinguish between the financial performance influenced by the market's fluctuating prices and that influenced by improvements in agricultural practices, which provide the optimal balance between economic viability and environmental sustainability. Reducing $CO_2$ emissions for a given level of economic return, or vice versa, can achieve this.

We have made some modifications to the decoupling technique and have discussed them in the context of decoupling methods for controlling energy output and ingestion. This work is a continuation of previous studies [53–55]. The economic exercise remains relevant despite decoupling, as previous research on decoupling strategies reveals a 1% annual change and a predicted change in decoupling. Given that the robustness and relationship have already been established through decoupling and subsequent portfolio turnover, which is a modification of research methodologies based on the decoupling methodology. The portfolio return has been estimated following the computing decoupling, and it is possible to predict the next five years and determine a 1% shift in $CO_2$ emissions using other indicators.This text begins with an examination of decoupling, followed by an estimation of portfolio return using weighted risk and return.

Finally, we discussed the fluctuation of this return. [54]. The negative portfolio return value represents the separated decoupling values. Consequently, for each category, the returns of the portfolio can be examined for a complete eight years [53]. The last step is to examine the variation in portfolio returns for every group. The next four steps make up the computation. Initially, we utilize beta decoupling to determine the average transformation of the environment, as well as the percentage change for each group individually. This indicated that the elastic range decoupling condition was satisfied. Initially, indicators are evaluated by multiplying the covariance and variance by 8/7 and 1%, respectively. Next, portfolio returns are calculated with high and low weighted risk by decoupling positive and negative attitudes. The weighted risk revealed that the weighted portfolio return encompassed both the total risk and the average return. Consequently, the highest and lowest attitudes of decoupling in this weighted risk model result in a 1% change in $CO_2$ emissions. This 1% change in decoupling has caused a 1% shift in the following areas: rice cultivation, enteric fermentation, field fire, managed soil, urea application, and manure management. The second step involves determining the average return and overall risk for each cohort after decoupling [56,57]. The mean return offers details about the conventional assessment, while the total risk indicates the standard deviation for each group over a period of eight years. Furthermore, the portfolio return with weighted risk is computed by adding the 100% risk premium to the adjusted return indicators of reduced environment, namely field fire, managed soil, urea application, manure management, enteric fermentation, and rice cultivation, after the average return and total risk [58,59]. A 1 percent change in the environment is then computed using the projected weighted return and risk of each cohort. In Section 3, the effects of the environment and the impact of a 1% change in

food security on $CO_2$ emissions degradation are also evaluated individually. Moreover, the decoupling shift at different junctures is ascertained by the groups [60]. For instance, a 1% shift in environment decoupling can be attributed to modifications in CLS-A, which encompass enteric in the field burning, managed soil, urea application, manure management, enteric fermentation, and rice cultivation [61].

### 3.2. Sources of the dataset

From 1992 to 2023, this study examines the $CO_2$ emissions implications of different agricultural techniques in Turkey, such as field burning, managed soil, urea application, manure management, enteric fermentation, and rice production, with the assistance of decoupling techniques. It then independently evaluates the portfolio returns using high and low weighted risk. The decoupling operation is divided into four distinct sections (A–D) for the purpose of analysis. Each subset represents a complete five-year period. Our Data in the World and the World Bank (WB) both have datasets that we analyzed using panel data (Appendix A). Inspired by eco-economic separation and a byproduct of techno-nationalism, accessibility is a determining element in the metrics' selection. The primary objective of this research is to assess the impact of CLS-A on the decoupling of $CO_2$ emissions in a statistically valid sample of Turkey. The significant increase in urea application in 2020 could be due to a combination of factors. These might include new government policies or subsidies that incentivized greater fertilizer use, an exceptionally favorable agricultural season that demanded more nutrients, or economic shifts such as a decrease in urea prices or an increase in crop prices making higher application more profitable. (Appendix B).

This increase is mostly attributable to the country's heavy use of coal to meet its growing energy demands. We investigate how these nations' actions affect decoupling on a global scale [62]. Based on their ecological situation and agricultural progress, we have ranked the nations [63]. It is widely believed that the price environment for commodities derived from these resources threatens sustainable economic growth due to their association with volatility. We determine that changes in the response variables are due to a decoupling estimate ranging from +1 to −2, which is a surrogate for measuring the effectiveness of agricultural initiatives to reduce $CO_2$ emissions harm [64]. We further explain the decoupling concept by examining the portfolio's performance. Conversely, if the decoupling analysis turns out to be unfavorable, we use techno-nationalism to rein in the $CO_2$ emissions decoupling effect and come up with workable solutions for the agricultural sector's transformation. The decoupling method was used to conduct a weighted calculation [65]. It demonstrates that as environmental conditions change by 1 percentage point, field burning, managed soil, urea application, manure management, enteric fermentation, and rice cultivation all change proportionately. The weighted values with a flexible range became readily apparent when distinct attitudes were uncoupled [66].

### 3.3. Portfolio returns

This study examines the efficiency of six key CLS-A indicators: managed soil, field burning, urea application, enteric fermentation, manure management, and rice cultivation. We calculate the portfolio return (average return and total risk) for each of them under three different aeration regimes. We categorize the aeration regimes into four distinct types: continuously flooded, single aeration, and multiple aeration in groups (A to D). Thus, we utilize current scientific knowledge about these activities to interpret the predicted environmental effects of $CO_2$ emissions, thereby establishing a connection between these indicators. Furthermore, the portfolio returns indicate things like yield, operational risk, profitability, weather-related hazards, and input cost variations. This is different from financial market indicators like decoupling or risk premiums, which show how investors perceive the risk and return in the capital markets as a whole. A new dataset that includes these financial metrics or a theoretical framework that explicitly links the market-level financial performance of agricultural ventures to their GHG emissions would be needed to set up a direct, empirically observed causal relationship between these particular financial market indicators and $CO_2$ emissions. For the first time, we have the data needed to assess the potential environmental impacts, risks, and economic feasibility of individual agricultural practices on individual farms. But when looking at the economy and finance as a whole, it is possible to speculate about a correlation between

financial indicators and $CO_2$ emissions; this is especially true when it comes to the explicit pricing of climate change risks on the market.

The CLS-A indicator's estimated weighted return value in the portfolio reveals how much of an impact the better decoupling approach has on the return. Two outcomes could occur. The first is the disparity between the variable's Turkish investment breadths. The indicators over the next eight years are a further factor to consider. The weighted risk and possible return are determined using the consequences of the environment, with the shared aid of decoupling. There are three steps to the calculation process [67]. The average return and risk value variable is one technical evaluation statistic that differentiates the four countries' economies (A–D) [55]. The average return is calculated in this research using total risk and average beta decoupling. That is consistent with the $CO_2$ emissions standard deviation for the past eight years. Secondly, we determined the portfolio weight by assigning a value to each of the three agricultural performance measures based on their relative relevance. We allot a certain percentage of the portfolio to each variable according to criteria like $CO_2$ emissions impact and economic development. The third phase relies on the concept of weighted rewards and risk [68]. This portfolio's enormous potential gains and losses are shown by the big swings. Additionally, the big swings highlight the disparity between risk and weighted return. To reduce $CO_2$ emissions risks to manageable levels, it is necessary to evaluate the viewpoints of the most risk-tolerant individuals, which can be achieved through changes in technology, changes in CLS-A, and strategic policies [69]. With this data, we were able to compute the average return and total risk, which we then adjusted using weighted percent. For the most part, previous studies have used a stochastic approach to explaining the portfolio selection problem, treating asset returns as random variables, and have assumed that the statistical feature of agricultural stock returns will remain constant [70]. Although decoupling can be advantageous, modeling portfolio selection under weighted return is problematic owing to the dynamic environment in different regions. In addition, multiply the total risk, average return, and estimated weighted value of each category by their corresponding values [71]. We calculate the weighted return value by averaging it. The risk value is an inevitable starting point for the weighted risk value. In this case, we estimate the total risk and average return as we approach the beta decoupling levels. It was able to accomplish this by taking a more solid and sincere approach to the percentage change in agricultural technology [70,72].

Pollutants released into the atmosphere by managed soil, field burning, urea application, enteric fermentation, manure management, and rice cultivation are the agricultural inputs that contribute to the degradation of naturally occurring soils. Industrial waste products, such as crop stubble and straw, are associated with emissions from field burning [73]. One major contributor to GHG emissions is the widespread use of urea fertilizer in CLS-A production. Among the numerous agricultural sources, such as manure management, field burning, urea application, enteric fermentation, and agricultural soils, among many others, urea—a nitrogen-based fertilizer—can release nitrous oxide, a potent CLS-A. Emissions originating from agricultural sources contribute to the overall $CO_2$ emissions [74]. The findings unequivocally demonstrate that emissions in each of the four categories have exhibited a progressively increasing trend over the course of the years. The emissions from managed soils have registered the most significant increase, which is most likely attributable to the increased application of fertilizers and manure to improve crop yields [75]. In addition, the data clearly demonstrate that CLS-A plays a crucial part in contributing to Turkey's overall $CO_2$ emissions profile. Around twenty percent of the nation's total emissions in 2020 came from the agriculture sector. The information presented in Appendix A provides a rundown of the $CO_2$ emissions that will be produced by Turkish CLS-A in the year 2020 [76]. One of the most significant contributors to $CO_2$ emissions within Turkish CLS-A is enteric fermentation, which accounts for approximately forty percent of the overall emissions [77]. Due to the country's robust economic growth, increasing incomes, population expansion, and reliance on $CO_2$ emissions, Turkey's overall GHG emissions have increased dramatically [78]. Furthermore, animal dung has the capacity to produce two substantial GHG gases, methane and nitrous oxide, during the manure management phases of storage, processing, and application. In Turkey, CLS-A soils account for more than 15% of the total $CO_2$ emissions. The production of nitrous oxide, the gas responsible for these emissions, is influenced by the cultivation of agricultural soil organic matter and the use of nitrogen fertilizers [79].

 

## 3.4. Portfolio equations

The weighted portfolio return was calculated for each agricultural management practice and strategy across different groups, considering four varying aeration conditions, in terms of both weighted risk (total risk) and weighted return (average return). Initially, weighted risk suggests a higher degree of unpredictability in the high- and low-risk zones. Similarly, the environmental impact and economic outcomes are subject to variation. When considering economic outcomes, the potential for deviations from the average return includes unstable yields and volatile profits. Additionally, the environmental impact—$CO_2$ emissions—is highly inconsistent across the various numbers of years in groups, which poses a high risk. Secondly, the agricultural strategies' weighted return outcomes are linked to both economic and environmental advantages. For example, a 1% modification in strategies results in an increase in profit, which in turn increases the return. The environmental strategies in terms of $CO_2$ emissions indicated that an average reduction and contraction in the environment are represented by a change of 1% in predictor variables. In the agricultural system, the average condition, such as continuous flooding and single and multiple aerations, is relevant in predictor variables that substantially influence $CO_2$ emissions. According to the decoupling approach, the portfolio return is estimated using the weighted average return and risk. These metrics are employed to assess the various agricultural strategies in relation to the degradation of climate change, including multiple aerations, single aerations, and consciously flooded ones. The individual indicators (managed soil, field burning, urea application, enteric fermentation, manure management, and rice cultivation) are analyzed using the weighted average return and individual examination of the total return and average risk in four distinct periods (A to D).

A comprehensive decision-making framework that considers both the reliability and environmental performance over time is required for evaluating various agricultural strategies for environmental mitigation, as indicated by the theoretical rationale for portfolio analysis. For example, accounting for environmental fluctuation and change in each group is necessary when comparing managed soil, field fire, urea application, enteric fermentation, manure management, and rice cultivation. Consequently, this investigation adopts the widely recognized portfolio return in terms of weighted average return, which reconceives the total risk and average return. Furthermore, we employ a rigorous technique to identify strategies of individual indicators in multiple aeration, single aeration, and consciously flooded conditions that offer both high environmental efficacy and low average $CO_2$ emissions, as well as high stability and low volatility. We also analyze the weighted average return and weighted risk. As a result, the most frequently employed formulations for weighted average return and risk in the context of portfolio return are as follows: The weighted average return is denoted by $R_{i,j}$ ($CO_2$ emissions) of agriculture strategy i under the condition of aeration, which is specified with $i$ over $n$ observed periods (Groups A-D) in Eq. 1. The weighted return specify with continuous flooding for managed soil, field fire, urea application, enteric fermentation, manure management, and rice cultivation is calculated as:

$$Weighted\ Average\ return_{i,j} = \frac{1}{n} \sum_{k=1}^{n} R_{i,j,k}$$

Eq.1

Eq. 2 illustrates the dispersion of individual outcomes in relation to the weighted average return, as denoted by

$$Weighted\ risk = \frac{1}{n \sum_{k=1}^{n} (R_{i,j,k} - Weighted\ Average\ return_{i,j})^2}$$

Eq.2

The most prevalent portfolio return formulations, which are calculated on the basis of decoupling, are the weighted average and weighted risk. The weighted risk in Eq.2 is typically represented by the σ of the return $R_{i,j,k}$, which is a measure of the dispersion of individual outcomes around the *Weighted Average return*$_{i,j}$. The agricultural strategy outcomes are displayed with the same value as i under the aeration condition j during a specific period "k", and the total number of periods is represented by "n". The individual effects of the individual predictor variables, such as managed soil, field burning, urea application, enteric fermentation, manure management, and rice cultivation, on the portfolio return have been demonstrated in terms of continuous, single, and multiple aeration.

## 4. Results

We used data for Turkey's nation from 1992 to 2023 and the portfolio return concept to look at how a resurgence with repeated and single aeration and continuous flooding affected high- and low-risk zones. The portfolio return is computed on the basis of the decoupling technique. Specifically, it is shown that a 1 percent change in both multiple and single aeration, as well as continuous flooding, results in a 1 percent change in CLS-A. In Fig 1, we can observe the pattern of wheat yields in Turkey throughout the four categories (A-D). With six different agricultural indicators, the risk and returns matrix has been evaluated to determine the economic benefits under four distinct groups (A to D) in rows and columns (Table 1).

The formatting condition for all CLS-A in this portfolio is evident in the returns, which feature three marks: a checkmark, an exclamation mark, and a heavy cross mark. The exclamation mark is recorded when the result is greater than 70%, and a hefty cross is indicated when the result is greater than 30%. The checkmark indicates a decoupling attitude of less than 70%. Furthermore, the marks indicate issues related to field burning, managed soil, and urban development within the CLS-A sectors [80,81]. The increase from 2.4 metric tons/ha in 2013–14 to 2.5 metric tons/ha in 2022–23 indicates that wheat yields have been rising consistently. We expect wheat yields to rise even further in 2023–24, reaching 2.7 tons per hectare. Fig 1 reveals that Turkey's wheat area remained virtually unchanged at 7,020,000 hectares (ha) between 2015/16 and 2019/20, compared to 7,860,000 ha in 2016/17. However, the wheat harvest has been more unpredictable, fluctuating between 15,000 metric tons in 2014–15 and 21,000 metric tons in 2017–18. The projected wheat harvest for 2023–24 is 19,500 million metric tons.

### 4.1. High-risk zone of CLS-A

According to Fig 1, the portfolio's CLS-A needs are assessed for multiple and single aeration and continuously flooded separately in high- and low-risk zones. Firstly, we analyzed nations with a high-risk factor. In our assessment, along with

| | | | Managed Soil | Field burning | Urea application | Enteric fermentation | Manure management | Rice cultivation |
|---|---|---|---|---|---|---|---|---|
| Continuously Flooded | Av.return | Group A | ✓ 0.164 | ✗ -0.511 | ! -0.190 | ✓ 0.130 | ! -0.090 | ✗ -0.105 |
| | | Group B | ✗ -0.197 | ✓ -0.047 | ✓ 0.052 | ✗ -0.007 | ✗ -0.526 | ✗ -0.215 |
| | | Group C | ✓ 0.202 | ✗ -0.394 | ✗ -0.396 | ✓ 0.099 | ✓ 0.204 | ✓ 0.154 |
| | | Group D | ✗ -0.135 | ✓ -0.106 | ✓ -0.073 | ✗ -0.004 | ✓ 0.096 | ✓ 0.042 |
| | Total Risk | Group A | ✓ 0.064 | ✓ -0.023 | ✗ -0.047 | ✓ 0.455 | ✓ 0.237 | ✗ -0.035 |
| | | Group B | ✓ 0.011 | ✓ 0.104 | ✓ 0.118 | ! 0.067 | ✗ 0.015 | ✓ 0.248 |
| | | Group C | ✓ -0.012 | ✗ -0.358 | ! 0.025 | ✓ 0.231 | ! 0.118 | ✗ 0.000 |
| | | Group D | ✗ -0.178 | ! -0.058 | ! 0.018 | ✗ -0.341 | ! 0.094 | ✗ -0.063 |
| Single Aeration | Av.return | Group A | ✗ -0.924 | ! 0.624 | ! 0.382 | ! 0.214 | ✓ 0.805 | ! 0.337 |
| | | Group B | ✓ 0.526 | ✗ 0.192 | ✗ 0.105 | ✗ 0.100 | ✓ 0.475 | ✗ 0.240 |
| | | Group C | ✓ 0.292 | ✓ 0.993 | ✓ 0.856 | ✓ 0.386 | ✗ -0.607 | ✗ 0.239 |
| | | Group D | ✗ -0.491 | ✗ 0.357 | ✗ 0.130 | ✗ 0.034 | ✓ 0.400 | ✓ 0.442 |
| | Total Risk | Group A | ! 0.045 | ! 0.055 | ✗ -0.119 | ✓ -0.105 | ✓ 0.066 | ✓ 0.090 |
| | | Group B | ! 0.122 | ✗ 0.000 | ✗ -0.127 | ✓ -0.157 | ✓ -0.003 | ! -0.171 |
| | | Group C | ✓ 0.373 | ✗ 0.047 | ✓ 0.074 | ✓ -0.125 | ✓ -0.014 | ✓ -0.030 |
| | | Group D | ✗ -0.553 | ✓ 0.147 | ✗ -0.177 | ✗ -0.318 | ✗ -0.178 | ✗ -0.360 |
| Multiple Aeration | Av.return | Group A | ✓ 0.331 | ✓ -0.273 | ✓ 0.129 | ✓ 0.139 | ✓ 0.556 | ✓ 0.401 |
| | | Group B | ✓ -0.045 | ✓ 0.055 | ✓ -0.119 | ✓ -0.105 | ✓ -0.066 | ✓ -0.090 |
| | | Group C | ✗ 0.292 | ✓ 0.993 | ✓ 0.856 | ✗ -0.386 | ✓ 0.607 | ✗ 0.239 |
| | | Group D | ✓ 0.491 | ✗ 0.357 | ✗ 0.130 | ✓ 0.034 | ✗ -0.400 | ✓ 0.442 |
| | Total Risk | Group A | ✗ -0.122 | ✓ 0.000 | ✗ -0.127 | ✗ -0.157 | ✓ 0.003 | ✗ -0.171 |
| | | Group B | ✓ 0.373 | ✗ -0.047 | ✓ 0.074 | ✓ -0.125 | ✗ -0.014 | ✓ -0.030 |
| | | Group C | ✓ 1.249 | ✓ 0.567 | ✗ 0.108 | ✗ -0.570 | ✓ 0.456 | ✓ 0.244 |
| | | Group D | ✗ 0.248 | ✗ 0.297 | ✓ 0.238 | ✓ 0.930 | ✗ 0.423 | ✗ 0.194 |

**Fig 1. Portfolio return of multiple and single aeration and continuously flooded in the high-risk zone.**

**Table 1. Structure of the indicator.**

| Structure of indicators | CLS-A indicator's and Emission |
|---|---|
| **Group (A to D), portfolio returns** | Three indications (continuously flooded, single aeration, and multiple aeration) were analyzed for overall risk and return and then divided into four groups based on specific risk scenarios. |
| **Portfolio return of rows** | The groupings in scenarios A, B, C, and D identify each hydrological indicator's data in terms of overall risk and returns. |
| **Portfolio return of columns** | The portfolio technique looks at environmental factors and agricultural surprises as unpredictable market risks, using data from six agricultural indicators, like managed soil-rice farming. |
| **Symbol** | The percentage change and coefficient of the indicator represent the magnitude of the effects of weighted average return and risk (total risk and average returns), with a checkmark (✓), red cross (✘), and exclamation mark (!). |

each country's efforts to combat climate change, we discovered that new technical regulations impact the CLS-A separation, characterized by three separate structural marks.

Similarly, in Fig 1, portfolio-weighted results of more than 70% are shown by the exclamation mark in high- and low-risk countries, respectively. The portfolio shows the formatting condition with a checkmark, exclamation point, and heavy cross mark. Checkmarks indicate less than 70% decoupling, exclamation marks more than 70%, and heavy crosses more than 30%. In terms of total risk and average return in weighted portfolio returns, the largest decoupling attitude is reported in Turkey (0.248% and 1.249%) and in field burning (0.567% and 0.297%) in groups D and C, with indicated heavy crosses, respectively. Similarly, rice cultivation shows the strongest attitudes with values of 0.194% and 0.244%, while enteric fermentation in single aeration has values of −0.318% and −125% in group C, as indicated by the heavy cross and checkmark. In the same scenario, the urea application in group D was 0.108%, and the corresponding value was 0.238. The managed soil decoupling attitudes show values of −0.553% for group C and 0.373% for group D during single aeration, represented by heavy cross and checkmark symbols, respectively.

The portfolio returns have been calculated using three dimensions: continuously inundated, single aeration, and multiple aeration. In groups C and D, the rise in cultivation is 0.154 and 0.042, respectively, with a checkmark in the average return. However, the total risk of continuously flooding is estimated to be 0.000 and −0.035, with a heavy cross, indicating the least decoupling. This demonstrates that a 1 percent change in continuous flooding results in a 1 percent change in CLS-A. In the same scenario, the single aeration result of rice cultivation indicated that the average return was 0.239 and 0.442 with a heavy cross and checkmark, and the total risk was −0.171 and −0.030 with an exclamation point and checkmark in groups B and C. It has been shown that a 1 percent change in single aeration results in a 1 percent change in CLS-A, which is comparatively higher than in continuously flooded.

**4.1.1. High-risk of continuously flooded.** Two scenarios have been interpreted to evaluate the performance of portfolio return (average return and total risk). First, the managed soil and enteric fermentation demonstrate the significant variation in weighted average returns across groups A to D, where the negative values (−0.038 and −0.467) respectively indicate high returns, signified by a prominent red cross in the managed soil category. Furthermore, the enteric fermentation shows a significant benefit in group D, with a 0.694 checkmark. Second, the overall yield risk is significantly affected by CLS-A, with the average return demonstrating positive impacts within the managed soil group (A, C, and D) and suggesting economic advantages as indicated by a check mark. The application of urea and rice cultivation presents significant risks, with values of 1.079 and 0.724, respectively, indicated by red cross checkmarks. Furthermore, the results of the managed soil also indicated that the highest decoupling change was in groups A and B, with values of 0.045 and −0.122, respectively, as indicated by the exclamation marks. This indicates that CLS-A exhibited the highest decoupling change. Hence, CLS-A from managed soils includes emissions resulting from the cultivation and management of agricultural areas, which encompass fertilizer applications, manure management, and other agricultural inputs. The

combustion of crop leftovers, such as straw and stubble, is associated with field burning emissions [82]. Use of urea fertilizer in agricultural cultivation is the source of urea application emissions. The nitrogen fertilizer urea can release nitrous oxide, a powerful $CO_2$ emission. Emissions from all agricultural activities, such as manure management, field burning, urea application, enteric fermentation, and agricultural soils, make up CLS-A's total emissions.

**4.1.2. High-risk of single aeration.** The average return of manure management is significantly influenced by CLS-A, with a coefficient of −0.016 (Group C). The overall risk of field burning and managed soil is also significantly elevated across all four groups, with values ranging from 1.043 to 1.390, and a checkmark value of 1.043. It indicated a substantial portfolio return within a singular aeration context. Regarding that Fig 2 demonstrates that growing rice releases a lot of $CO_2$ emissions. The projected 2020 $CO_2$ equivalent emissions from rice cultivation were 261.53 million tons. Overall, this represents approximately 1.4% of the global $CO_2$ emissions. Compared to intermittent flooding practices, persistent flooding in rice cultivation results in higher $CO_2$ emissions [83]. This is because, in anaerobic conditions, methane is more readily produced in continually flooded rice paddies, making them a powerful source of $CO_2$ emissions [84].

The reliance on energy-intensive agriculture and GHG emissions continues to be significant; additionally, the relationship between agricultural exports and environmental sustainability creates a unique context for analysis, leading to notable growth of farming sectors with minimal risk to these exports. However, this rapid expansion has directly impacted

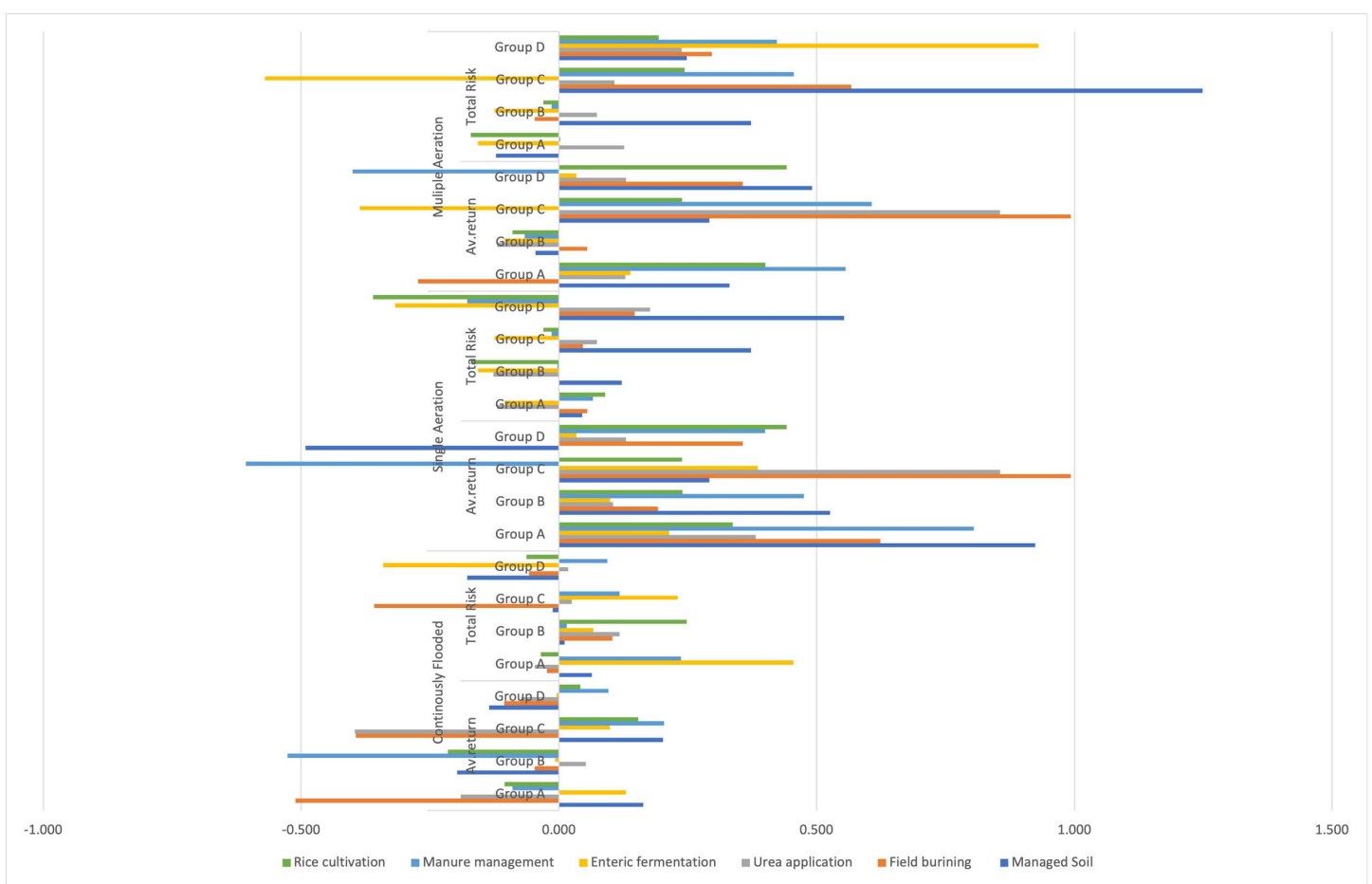

**Fig 2. Portfolio return of multiple and single aeration and continuously flooded in high-risk.**

the environment and contributed to increased GHG emissions. Turkey relies substantially on energy from many sources; hence, GHG emissions in Turkey have significantly increased [78]. Furthermore, emissions resulting from rice cultivation have increased over the years. Emissions from rice cultivation have increased by approximately 50% since 1990. Several factors have contributed to this, including the rise in fertilizer use, the expansion of rice agriculture into previously uncultivated areas (such as wetlands), and the overall increase in rice production [85].

**4.1.3. High-Risk of multiple aeration.** The average return of managed soil is markedly affected by CLS-A within groups C and D. Furthermore, the overall risk associated with field burning and managed soil is also considerably higher across Groups C and D, with values ranging from 1.169 to −0.810, accompanied by an exclamation mark, indicating a negative impact on portfolio return. In the most recent urea application, the field fire exhibited a high-risk indication, marked with a red cross, which suggests the need to manage the portfolio risk associated with these challenges—the negative returns of −0.616 and −0.810 for manure management point to the economic impact on CLS-A. With the help of a variety of agricultural indicators across four hydrological groups (A-D), this portfolio return offered a stochastic evaluation of the system's effectiveness in terms of economic efficiency in Turkey. There is a high probability that a heterogeneous parametric environment (CLS-A) occurs when portfolio returns exhibit systematic volatility or negative extensivity. There is a significant gap between systemic risk and unjustified economic returns, which is the primary strategic implication that this portfolio return study reveals.

Due to the tendency for portfolio risk in highly influential locations, risk mitigation priorities in Turkey should be adjusted and updated with an operational assessment across all regimes. This is because field burning and urea application are two of the most important risk mitigation strategies. Second, the value of sustainable measures of negative average return examined the value of manure management in various aeration acts as a powerful economic argument in CLS-A, which is against it before we consider environmental issues. Furthermore, this argument is against it before we consider environmental concerns. In the third place, there should be a strategic focus on recreating the agricultural scenario that resulted in the elevated risk, such as enteric fermentation in a consistently flooded environment.

## 4.2. Low-risk zone of CLS-A

Fig 3 illustrates the portfolio return of low-risk countries weighted by risk and return with varying ratios in the CLS-A sector. Initially, the approximated results are analyzed for both single and multiple aerations and continuous flooding in the low-risk zone.

The maximum weighted risk in group D is estimated to be field burning (1.390), multiple aeration (0.693), and manure management (1.079) in China, a high-risk country [86].

From about 70% in 2000 to about 75% in 2019, the percentage of CLS-A in Turkey's water abstractions has been steadily rising. These data points to the fact that CLS-A is using more and more water in Turkey. Additionally, Turkey's irrigated land share has been steadily rising, from about 20% in 2000 to about 25% in 2019. This indicates that irrigation is becoming more important for Turkish farmers when it comes to crop production [87]. Throughout the years, CLS-A's contribution to Turkey's $CO_2$ emissions has remained around 20%. Therefore, CLS-A probably isn't a big deal when it comes to Turkey's $CO_2$ emissions [88]. From about 10% in 2000 to about 8% in 2019, CLS-A's proportion of Turkey's overall energy consumption has been steadily declining. These data point to an increase in CLS-A's energy efficiency in Turkey. It has been more common to take phosphorus from the soil in Turkey than to add it, resulting in a negative phosphorus balance since the year 2000.

**4.2.1. Low-risk zone of continuously flooded.** Within the same scenario of a high-risk zone, the persistent flooding in the low zone is indicative of economic productivity as well as the implications of environmental degradation with various agricultural indicators. The enteric fermentation exhibits the maximum intensity of average return with a positive attitude in Groups A to D with checkmark. This is the case in the portfolio return. On the other hand, the controlled soil, rice cultivation, and manure management have all demonstrated negative returns, particularly in Group B with low intensity.

| | | Managed Soil | | Field burning | | Urea application | | Enteric fermentation | | Manure management | | Rice cultivation | |
|---|---|---|---|---|---|---|---|---|---|---|---|---|---|
| **Continuously Flooded** | **Av.return** | ✓ | 0.038 | ✗ | -0.058 | ✗ | -0.042 | ✓ | 0.694 | ! | -0.280 | ✗ | -0.111 |
| | | ! | -0.203 | ✓ | 0.277 | ✗ | 0.097 | ! | -0.045 | ! | -0.036 | ! | -0.043 |
| | | ✓ | 0.126 | ✗ | 0.045 | ✗ | 0.023 | ✗ | -0.479 | ✗ | -0.574 | ✗ | -0.206 |
| | | ✗ | -0.467 | ! | 0.108 | ✓ | 0.478 | ✗ | -0.663 | ✓ | 0.264 | ✓ | 0.255 |
| | **Total Risk** | ✗ | 0.278 | ✗ | 0.414 | ✗ | 0.141 | ✓ | 1.832 | ! | 0.424 | ✗ | 0.178 |
| | | ✓ | 0.708 | ✓ | 0.742 | ✗ | 0.168 | ✗ | 0.052 | ✗ | 0.217 | ✗ | 0.175 |
| | | ✓ | 0.832 | ✗ | 0.326 | ✗ | 0.200 | ✓ | 1.356 | ✓ | 0.780 | ✗ | 0.254 |
| | | ✓ | 0.834 | ✓ | 0.891 | ✓ | 1.079 | ! | 0.668 | ! | 0.525 | ✓ | 0.434 |
| **Single Aeration** | **Av.return** | ✓ | 0.187 | ✗ | 0.023 | ✓ | 0.141 | ✗ | -0.200 | ✓ | -0.353 | ! | -0.226 |
| | | ✗ | -0.401 | ✓ | 0.504 | ✓ | 0.123 | ! | 0.004 | ✓ | -0.347 | ✗ | -0.319 |
| | | ! | -0.136 | ✗ | -0.085 | ✗ | -0.053 | ✗ | -0.317 | ✗ | -0.616 | ✓ | -0.114 |
| | | ! | -0.021 | ! | 0.125 | ✗ | -0.011 | ✓ | 0.427 | ✓ | -0.337 | ✗ | -0.269 |
| | **Total Risk** | ✗ | 0.402 | ✗ | 0.448 | ✓ | 0.566 | ! | 0.813 | ✗ | 0.355 | ✗ | 0.261 |
| | | ✓ | 1.043 | ✓ | 1.344 | ! | 0.288 | ✗ | 0.081 | ✓ | 0.771 | ✓ | 0.743 |
| | | ✗ | 0.159 | ✗ | 0.107 | ✗ | 0.054 | ! | 0.774 | ✓ | 0.800 | ✓ | 0.595 |
| | | ! | 0.589 | ✓ | 1.390 | ✓ | 0.405 | ✓ | 1.540 | ✗ | 0.415 | ✗ | 0.392 |
| **Multiple Aeration** | **Av.return** | ✓ | -0.026 | ✗ | -0.959 | ✓ | 0.110 | ! | 0.237 | ✓ | 1.006 | ! | 0.178 |
| | | ✓ | -0.047 | ✓ | -0.028 | ✗ | -0.028 | ✓ | 0.622 | ! | -0.012 | ✓ | 0.486 |
| | | ! | -0.186 | ✓ | -0.167 | ✗ | -0.086 | ✗ | -0.085 | ! | 0.064 | ✗ | -0.217 |
| | | ✗ | -0.305 | ✓ | -0.004 | ✓ | 0.162 | ✗ | -0.252 | ✗ | -0.810 | ! | 0.039 |
| | **Total Risk** | ✗ | 0.231 | ✓ | 1.169 | ✓ | 0.366 | ! | 0.680 | ! | 1.826 | ! | 0.614 |
| | | ✗ | 0.139 | ✗ | 0.275 | ✗ | 0.076 | ✓ | 1.154 | ✗ | 0.318 | ✓ | 0.899 |
| | | ✓ | 0.791 | ✗ | 0.490 | ! | 0.245 | ✗ | 0.566 | ✗ | 0.216 | ✗ | 0.287 |
| | | ✓ | 0.751 | ! | 0.693 | ! | 0.230 | ✗ | 0.393 | ✓ | 3.079 | ✗ | 0.225 |

**Fig 3. Portfolio return of multiple and single aeration and continuous flooding in a low-risk zone.**

With an exclamation mark, the total risk value regarding rice cultivation and field burning significantly demonstrates a high risk, particularly with regard to the application of urea and the management of manure.

It appears that the soils of Turkey are losing their phosphorus content. Since the year 2000, Turkey has also had a negative nitrogen balance, which means that the country is removing more nitrogen from the soil than it is adding. This indicates that the nitrogen levels in Turkey's soils are declining. These conclusions indicate that CLS-A in Turkey is facing difficulties related to water scarcity, soil degradation, and climate change. The Turkish government must take action to address these concerns if CLS-A continues to operate in the future. Consequently, the utilization of rice and urea has resulted in a 0.392 and 0.405 reduction, respectively, in agricultural productivity within group D. In the low-risk zone, group C has the highest weighted risk at 0.774, which is 1.98 times higher than group D's 1.540. Since 1990, when it was 37.9 million hectares, the irrigated area for growing rice has grown significantly, reaching 125.4 million hectares in 2020 [89]. From 1990 to 2020, the estimated emissions from rice agriculture have climbed continuously, reaching 261.530 million tons of $CO_2$ equivalent. Fig 4 demonstrates that rice cultivation has the greatest attitude in single aeration, which is vital because rice is both a staple crop for many people's diets and a big agricultural crop in the country. Turkish rice yields have steadily increased over the previous decade, rising from a respectable 7.6 metric tons per hectare in 2013–14 to an impressive 8.30 metric tons per hectare in 2023–24.

**4.2.2. Low-Risk Zone of Single Aeration.** In the single aeration, the average return shows a highly volatile and negative attitude in manure management and managed soil in Group C. However, field burning and urea application surprisingly show comparatively low portfolio risk but the positive attitude of urea application and enteric fermentation are significantly influenced by environmental degradation in Group D. Furthermore, field burning and managed soil exhibit the highest total risk in Groups A and B with an exclamation mark. Furthermore, regarding CLS-A there are three major reasons for this impressive expansion. First, let's discuss the technological advancements [84]. Modern seed types,

**Fig 4. Portfolio return of multiple and single aeration and continuously flooded.**

precision CLS-A methods, and integrated pest control concepts are just a few examples of how Turkish farmers are fully embracing cutting-edge agricultural technology [90]. There is no denying that these technological advancements have been the foundation of the increased rice harvests. Secondly, backing from the government: The Turkish government has done an admirable job of becoming a reliable backer of rice farming, offering crucial assistance in the form of subsidies, loans, and outreach programs. And with strategic backing, the nation's rice farming has become much more efficient and productive [91]. Thirdly, beneficial climatic conditions have emerged; recently, Turkey's climate has been highly favorable for rice cultivation.

The positive dynamics of $CO_2$ emissions have created an enabling environment for the enhancement of rice yields. Fig 5 illustrates that under group D, the Turkish economy and agricultural landscape have benefited greatly from the increasing rice yields achieved through enteric fermentation, manure management, and rice planting. Increased yields allow farmers to extract more rice from their land, which improves their financial situation. Additionally, Turkish rice gains a competitive edge on the world stage due to a dramatic decrease in production overheads brought about by an increase in yields.

**4.2.3. Low-Risk Zone of Multiple aeration.** The average return across multiple aeration practices demonstrates a negative trend in Group D, particularly in areas such as manure management, field burning, and managed soil, as indicated by the red cross, signifying a detrimental impact on environmental indicators. However, the development and

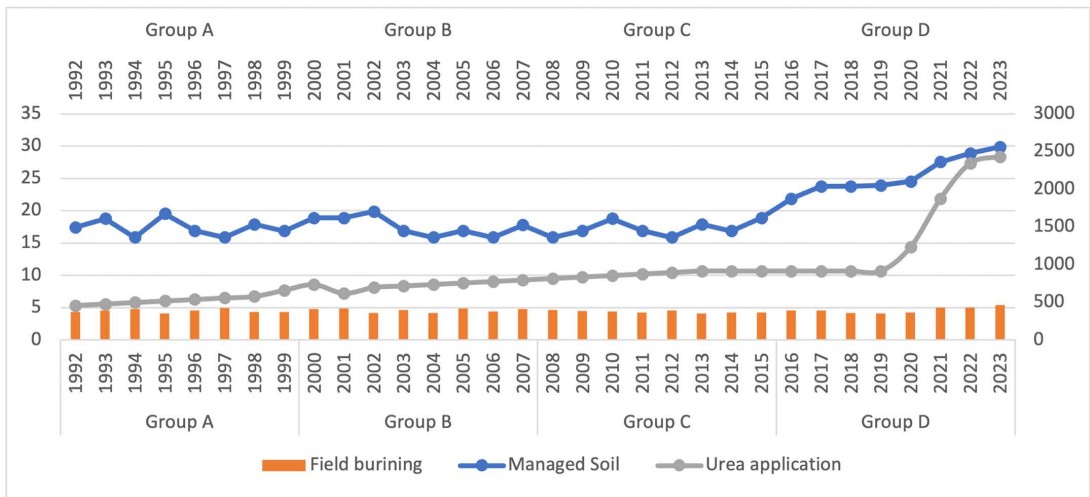

**Fig 5. Managed Soil, Field Burning, and Urea Application.**

enforcement of agricultural policies can help maintain and mitigate this gap. On the other side, the total risk of field burning shows the highest attitude in Group A, with an exclamation mark for manure management and rice cultivation in Turkey. Like the rest of the world, Turkey is facing the daunting prospect of climate change as it pertains to agricultural output. Farming community may have to adjust to changing weather conditions and look into new agricultural technology as a safety net if we want to survive this enormous challenge and keep or increase our rice harvests for the future. Agricultural activities such as liming, field burning, and rice planting also contribute to emissions.

Additionally, it emphasizes the significant role of the agricultural sector in Turkey's $CO_2$ emissions profile, representing approximately 20.21 percent of total emissions, allowing us to infer several critical insights. The critical importance of the Turkish agricultural industry reducing its $CO_2$ emissions immediately [92,93]. To lower Turkey's total carbon footprint and achieve its goals for mitigating climate change, this step is crucial. Measures to decrease emissions: the agricultural sector in Turkey can decrease its $CO_2$ emissions by implementing several proven solutions. Improving manure management, reducing nitrogen fertilizer use, and switching to more environmentally friendly farming methods are all part of these steps [73]. From 2014 to 2023, the line graph in Fig 6 displays the maize yields in Turkey. Crop yields for maize have been trending upwards for a while now. This trend is a response to climate change and aims to secure a future food supply, which ranks somewhat higher in group D for growing rice. Rising rice harvests in Turkey are an encouraging trend [73]. Improving efficiency and output while capitalizing on favorable weather conditions is a hallmark of the remarkable transition that the country's rice farmers are presently experiencing. But we must not lose sight of the fact that increasing rice yields may not be sustainable forever [74].

Additionally, the diverse agricultural practices under varying aeration conditions demonstrate the sustainable development of Turkey's agricultural sector in terms of portfolio return. This development has the potential to significantly enhance food security and reduce the sector's $CO_2$ emissions impact in terms of portfolio return. The CLS-A need is assessed for multiple and single aeration and continuously flooded separately in high- and low-risk zones with average and total risk. The portfolio returns the value of soil and agriculture, as indicated by the estimated results. Additionally, the portfolio return has the potential to promote the development of productive, sustainable, and diverse agricultural practices in Turkey by enabling producers to make decisions that are more in accordance with market forces, thereby improving food security [94]. The estimated results are also consistent with the stated objectives, which promote sustainable methods and fruitful cultivation, as well as the reduction of environmental degradation caused by $CO_2$ emissions in Turkey. The agriculture

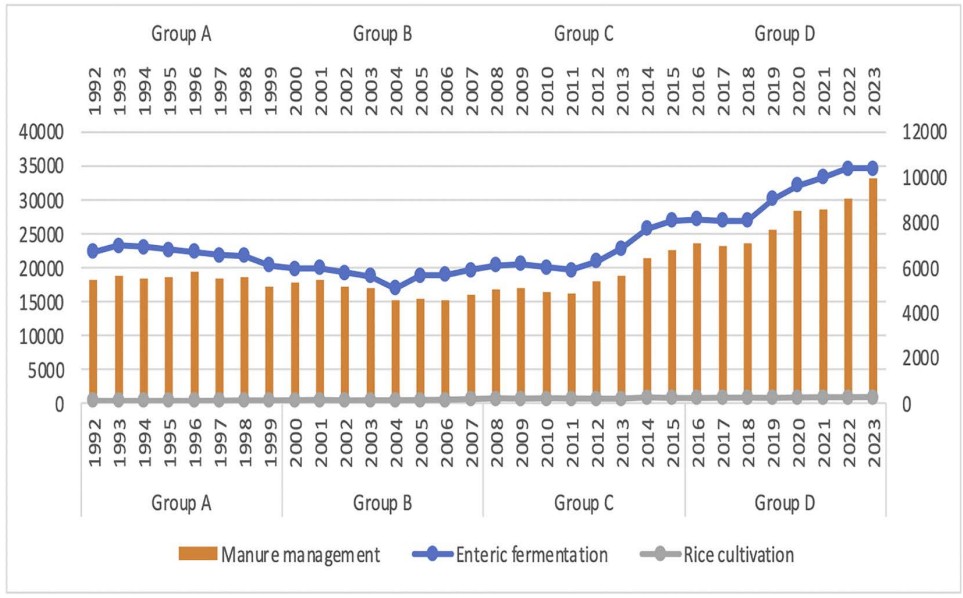

**Fig 6. Enteric fermentation, manure management, and rice cultivation.**

sector is a high-potential source and sink of GHGs, including $CO_2$ emissions, accounting for approximately 24% of total anthropogenic emissions in terms of land management, soil management, field fires, rice cultivation, and urea application. Consequently, it is imperative to examine the dynamic and structural aspects of agricultural emissions [95]. Additionally, the changes in agricultural land use are directly and indirectly associated with various sectors of GHG emissions, including the management of soil, field fire, urea application, enteric fermentation, manure management, and rice cultivation. A variety of spatiotemporal factors, including seasonal fluctuations and local practices, influence Turkey's land-use emissions. Consequently, the primary concern is the stabilization of agricultural emissions, which is crucial for the mitigation of environmental change and the preservation of food security [96]. Additionally, the practice of enteric fermentation, urea application, and manure management directly correlates with GHG emissions, providing an empirical foundation for evaluating the potential of a unique method to significantly contribute $CO_2$ emissions. As a result, the technological revolution and changes in agricultural policy are making emissions happen more slowly. Ultimately, these practices yield positive returns while reducing risk, providing dynamic evidence to inform sustainable policies that encourage producers to adopt more market-aligned methods and promote sustainable development, thereby contributing to both the goals of emission reduction and food security.

## 5. Discussion

We are observing that the frequency of droughts, floods, and heat waves is increasing, and they are inflicting greater damage with each occurrence. It is presenting producers with additional obstacles in the areas of animal husbandry and plantation. The data depicted in the image is unreliable for scientific analysis due to the alterations. It is essential to have access to the most up-to-date data on CLS-A to make well-informed decisions concerning agriculture. [97]. Adopting CLS-A practices is necessary to mitigate the consequences of climate change on food security and nutrition [47,98]. These methods facilitate the integration of food production and climate change adaptation, thereby providing a means to enhance food security [99]. Furthermore, $CO_2$ emissions are predominantly affected by agricultural factors, as evidenced by the estimated results. It indicates that the impacts of agricultural factors, including continuous flooding, monoculture,

and various aeration methods, as measured by specific indicators, are likely connected to environmental sustainability practices [100]. Agricultural practices require human involvement, which means that the development techniques used in the agriculture sector can help control GHG emissions. However, the utilization of agricultural factors is observed to result in greater environmental degradation in terms of $CO_2$ emissions [101]. Moreover, it is crucial to ascertain the actual situation of the agricultural factor, which contributes over 6% to Turkey's GDP, and the country's commitment to environmental sustainability, based on the high- and low-risk zones of CLS-A. Consequently, this study investigates the environmental challenges that may potentially arise from agricultural factors and create environmental pressure in terms of $CO_2$ emissions [102].

In terms of high-risk zones of CLS-A, this portfolio's return is analyzed within the matrix across three fundamental dimensions of CLS-A and environmental impacts for agricultural management and policy formulation. These dimensions encompass environmental, governance, and social aspects. To begin, there is an imbalance in the weighted Av.retun and risk between the undisturbed systemic risk of agricultural problems and the economic return. Regarding Rice Cultivation and Managed Soil, the overall risk demonstrates a high level of intensity in a single aeration setting. Because of this, it is imperative that decisions on investments not be focused exclusively on the Av.return of the company. While return is important, the weighted method of the overall risk component is necessary. As an additional point of interest, the high risk of urea application and field burning in various groups revealed that these activities indicated an unacceptable level of fragility. Regulations should be amended, and stringent measures should be taken to minimize these behaviors, since the potential cost of failure associated with them is greater than the marginal economic gain that can be achieved with modified CLS-A regulations. The second consideration is contextual optimization and the dependency of managed soil and single aeration conditions. This is because both of these conditions are greatly influenced by the environment when continuous flooding is present. In addition, the identification of the niche environment for enteric fermentation is expected to be advantageous hydrological dynamics that directly enhance economic yield while simultaneously minimizing risk. The third point is that the policy intervention of CLS-A concerning portfolio return updates and revisions in Groups C and D, which encompasses continuous, single, and repeated aeration, has to modify the systemic stability goal and sustainable agriculture in light of the degradation of the environment.

The CLS-A program presents a critical opportunity to improve the fortunes of smallholder farmers in Turkey, including limited land and the effects of climate change. Some of the most significant CLS-A strategies are the promotion of regenerative agriculture, crop diversification, biochar application, and agroforestry, all of which enhance soil health and decrease GHG emissions [35,103]. The nutritional value and production of crops are both affected by the state of the soil, which is essential to good crop nutrition. We can help reverse the declining trends in food quality by giving farmers climate-resilient technology and sustainable practices like regenerative agriculture and precision agriculture [104]. Household incomes can be increased, self-sufficiency in food production can be improved, and food security can be strengthened because of the increased agricultural productivity that CLS-A has achieved. CLS-A practices are more likely to result in improved food consumption, increased dietary diversity, and enhanced overall food security in households [104].

Additionally, the food supply and food markets are at risk due to climate change; however, these risks can be mitigated by enhancing producers' adaptive capacity and improving resource use efficiency [105]. In Turkey, managed soil is mostly linked to the sources of $CO_2$ emissions, particularly agricultural operations. Like, traditional tillage emits GHG emissions. Sustainable approaches, such as zero tillage, are currently being studied to eliminate these sources of emissions [106–108]. The major goal is to improve soil management for emissions through carbon sequestration and to minimize the overall $CO_2$ footprint in the agricultural sector [109]. Secondly, the aforementioned estimated results also indicated that the field burning of the agriculture sector in the country contributes to GHG emissions, specifically $CO_2$ emissions. The practice has diminished, and the application of urea constitutes an independent source of these emissions. However, the precise factor may differ depending on the season and soil conditions. The rise in Turkey from 1990 to 2020 was predominantly influenced by $CO_2$ emissions, manure management, and the overall greenhouse gas emissions within the country

[110]. Third, the total risk of enteric fermentation in Turkey is a substantial contributor to GHG emissions in the agricultural sector in 2023. This includes an estimated 71.8 Mt $CO_2$ equivalent, which represents a 38.4% increase since 1990. Consequently, the anticipated outcomes of this investigation also demonstrated its substantial influence on groups C and D [111]. Fourth, both manure management and rice cultivation are significant contributors to $CO_2$ emissions equivalent emissions, which represent the largest component of overall greenhouse gas emissions from agricultural activities and the energy sector [112]. Additionally, manure management was one of the largest sources of GHG emissions in Turkey in 2020, along with livestock. Sustainability practices can mitigate the substantial impact of emissions from the agriculture sector on environmental degradation [113]. Additionally, it is imperative to adopt an integrated strategy for mitigating climate change, which involves collaboration at the local, national, and international levels. Governments and stakeholders must implement comprehensive economic, social, and environmental strategies to enhance climate risk management, mitigate vulnerability, and enhance resilience [114]. Consequently, farmers have been unable to diversify their livestock and crop cultivation options. The modifications have rendered the information in the image unsuitable for scientific research [115]. Accurate and up-to-date information regarding CLS-A should be the foundation for any modifications to agriculture policy and practice [116]. Turkey is a prominent producer of wheat, and it is a staple in the cuisines of numerous individuals worldwide [117]. Consequently, producers are applying an increased quantity of fertilizers to the soil [118].

In terms of low-risk zones of CLS-A, the continued flooding in Turkey indicated that enteric fermentation and agricultural soil are the largest sources of non-GHG emissions in the Turkish agricultural sector. This implies that these mitigation actions could yield significant benefits to their stability, although the outcome may be highly uncertain due to factors such as feed quality, market volatility, and seasonal changes. The low-risk zone is characterized by conditions consisting of Turkey. The results suggested that in Group A, there was a high potential for carbon sequestration and environmental degradation. This was in reference to the managed soil. In addition, it is well known that rice farming changes the carbon content of the soil in Turkey, but non-continuous flooding helps limit the amount of $CO_2$ emissions caused by the application and adjustment of fertilizer. It is also important to note that the management technique aligns low risk and high return for both managed soil and organic fermentation with a reliable foundation of fulfilling national targets with excellent confidence and environmental security.

Furthermore, the management techniques exhibit the least advantageous situation in terms of single aeration. This is due to the prevalent negative attitude of Av.return and the high total risk, both of which serve as key issues in the region of agriculture. Consequently, emissions and reductions in agricultural practices, such as enteric fermentation and controlled soil in Turkey, are efficient in reducing emissions. Regarding multiple aeration, elevated risk and complexity exhibit a significantly challenging pattern of negative average returns across numerous strategies. While the revised method seeks to optimally balance emissions and potential, it is implemented across Turkey's diverse environmental zones and potential yields. Meanwhile, potential stable strategies within the agriculture sector aim to prevent catastrophic emissions.

Farmers' adaptive capacity and resource utilization efficiency can be improved to mitigate the risks associated with climate change, which disrupts food markets and threatens the food supply [98]. An integrated strategy that involves collaboration at the local, national, and international levels is critically important for mitigating climate change in Turkey [99]. The implementation of comprehensive economic, social, and environmental strategies by governments and stakeholders is essential for the enhancement of climate risk management, the reduction of vulnerability, and the increase of resilience [119]. In this context, the relationship may be interpreted as both a reactive mechanism and a recently adopted policy. The rise in pollution levels compels Turkey's government to adopt policies and social measures, including enhanced investments in agriculture, clean energy technologies, and incentives to promote renewable integration [120]. Furthermore, increased $CO_2$ emissions in high-risk zones serve as a warning sign for environmental degradation, which in turn prompts the development of strategies for mitigation and adaptation, particularly in the agricultural sector, characterized by growing environmental awareness and international climate commitments [121]. Climate-resilient pathways should prioritize evidence-based solutions, strengthen local institutions, harmonize agricultural and climate policies, and integrate climate

and agrarian financing, as noted by the work [108,122]. By emphasizing adaptable and context-driven solutions, this integrated approach is distinguished by its innovative policies and financing mechanisms [123].

The estimated results clearly indicate that CLS-A practices are correlated with the most positive food security outcomes within the highest-risk, most vulnerable zone. This suggests that the adaptation and modification strategy is effective. [124]. Additionally, the high zone portfolio return directly influences GHG emissions; therefore, food security and resilience have significantly contributed to environmental changes [125]. In terms of managed soil, field burning, urea application, enteric fermentation, manure management, and rice cultivation, this study provides policymakers with the information they need to make strategic changes in the agricultural sector [126]. Furthermore, it highlights the empirical evidence necessary for advancing food security, agriculture, and the optimal allocation of resources in Turkey. Numerous Turkish farmers are presently utilizing diverse methods, such as crop rotation, to preserve soil health and mitigate excessive water consumption [127,128]. An additional contributing factor to Turkey's unprecedented wheat harvest was the favorable weather conditions [129]. Turkey has recently encountered mild winters and an abundance of rainfall in the course of the growing season [130]. That is why growing wheat yielded such excellent results. [131]. An escalation in wheat production signifies a positive development for Turkey, particularly in light of the global demand for food. Turkey's reliance on imports will decrease as its wheat production expands [132]. The Turkish government should guarantee that producers continue to receive support payments to assist them in enhancing their wheat harvest [133].

## 6. Conclusion

Turkey can reduce maize harvest risks by taking the following steps: The CLS-A sector can improve collaboration among many agencies, strengthen the important ministry of food, CLS-A, and livestock components, and take other positive actions to prevent climate change. The CLS-A increases productivity, mitigates emissions, improves climate resilience, and the agricultural sector in Turkey is currently experiencing real effects from climate change, changes in farming practices, and uneven development, which have caused a rise in environmental costs, including GHG emissions. The decoupling process consequently integrates sustainable agriculture products into this portfolio. We intend to disseminate information regarding the possible effects of climate change on CLS-A and the potential measures that can be implemented to mitigate the risks.

This study offers a fundamental empirical contribution by presenting quantitative evidence of a significant, non-linear relationship between the adoption of CLS-A practices and a subsequent reduction in $CO_2$ emissions within the agricultural sector in Turkey, thereby demonstrating the efficacy of CLS-A. Furthermore, in terms of CLS-A, the degradation of naturally occurring soils is facilitated by the agricultural input of pollutants discharged into the atmosphere by managed soil, field burning, urea application, enteric fermentation, manure management, and rice cultivation. Crop residue and straw, which are industrial waste products, are linked to emissions from field burning. Our primary objective is to enhance interagency collaboration and develop capabilities. Secondly, we aim to improve the capacity to manage drought. The objective underscores the necessity of enhancing the capacity of the committees and agencies that are responsible for the administration of agricultural droughts. The findings also have important policy implications for Turkey's long-term transition to CLS-A. For this to happen, resources should be strategically shifted from conventional input subsidies to subsidized adoption of proven CLS-A techniques that improve $CO_2$ emissions sequestration and decoupling. A third point is that there are training and research programs in place to address the impacts of local and regional climate change. It is highly recommended that the ministries of food, CLS-A, and cattle, in conjunction with their affiliated institutions, establish a special department within their respective provincial organizations to address climate change-related issues.

To adapt to the CLS-A sector's impacts of climate change, it is crucial to coordinate Turkey's policies, strategies, and legislative framework, as stated in the objective. The goal of these programs is to promote sustainable farming practices while simultaneously ensuring the safety and security of our food supply. Incorporating climate change considerations into existing policies and initiatives is crucial for effectively addressing these issues. Second, it highlights the importance of

conducting and expanding scientific studies and research to understand the impacts of climate change on CLS-A and to facilitate adaptation to these changes. The agricultural sector should increase its utilization of biofuels to mitigate environmental contamination. Furthermore, it is imperative to implement initiatives that prioritize financial incentives for organic agriculture, sustainable agricultural practices, and land conservation. Although the agricultural sector must decrease its reliance on fossil fuels and increase its utilization of renewable energy sources, the implementation of alternative renewable energy production methods, particularly hydroelectric systems, may result in environmental issues. This empirical finding indicates that the adoption of CLS-A practice, particularly in the form of high- and low-risk zones, has a statistically significant value in decoupling. The lowest value is observed in the low-risk zone for enteric fermentation, soil, and urea application, in comparison to manure management and rice cultivation. The main policy recommendation is that the decoupling results indicate Turkey should subsidize and incentivize the transition to modified CLS-A practices in both public and private sector land management to alleviate the environmental pressure from GHG emissions.

The paper focuses on the following main points. To begin, economic development goals are a common impetus for agricultural research and development projects, which have traditionally prioritized increasing production. This has led to less focus on how to adjust to a changing climate. Second, CLS-A's shift to climate revolution adaptation: The goal emphasizes the need to redirect research efforts towards adapting to climate change, especially those that focus on soil and water resource protection. Developing irrigation systems, adjusting production patterns, increasing crop diversity, and disaster management strategies are all necessary to address the impacts of climate change on CLS-A on a national scale, especially in relation to drought. The third component of the plan is socio-economic research. This study will focus on CLS-A, food, the environment, and rural development on a national scale. It will take climate change into account. The findings of this study will pave the way for innovative policies that encourage the expansion of the country's agricultural sector in response to climate change, and they will also assist in securing the livelihoods of specific communities, such as women farmers. The fourth aspect is ensuring long-term food security. The plan stresses building a food security system around data uncovered by R&D initiatives that account for the impacts of climate change. Aiming to meet customer expectations while responding to changing factors affecting $CO_2$ emissions, this system is constantly evolving. The fifth objective, which is to build a database and information system, emphasizes the importance of developing this system at a regional or basin scale, or even nationally. Lastly, the scale effect is valid in Turkey, as evidenced by a positive economic globalization coefficient. We should promote foreign direct investment in environmentally favorable and technology-intensive practices, as these investments have the potential to contribute to economic growth and employment. Additionally, the application of several environmental standards in international business can contribute to the mitigation of environmental contamination.

As a result, the identification of appropriate CLS-A strategies for the reduction of GHG emissions and the preservation of crop productivity necessitates a thorough examination of local environmental factors, such as soil condition, climate, and their interactions with alternative management systems. Various sociocultural, economic, and demographic factors influence the adoption of CLS-A. Although practices such as crop rotation, integrated soil fertility management, crop diversification, and intercropping are frequently implemented, their widespread adoption remains low as a result of limitations associated with farm size, infrastructure, equipment, machinery, irrigation access, extension services, and obtaining weather information. The adoption of CLS-A can be improved by providing producers with the necessary inputs, providing context-specific CLS-A technologies and training, and enhancing extension programs, as recommended by this study. The successful implementation of CLS-A practices can be substantially impeded by a lack of farmer awareness and expertise, as training is essential for connecting farmers with supportive institutions, reliable input suppliers, and access to credit facilities. A significant direction for future research pertains to the cross-country validation of our established decoupling model, including its application in Turkey. And using the different agricultural economies with the CLS-A and $CO_2$ emissions link will make the sustainable CLS-A more important for global policy. Furthermore, this method can be revised regularly to accommodate evolving climatic effects; it will facilitate the assessment and monitoring of the impact of climate

change on agriculture. The future research should concentrate on the development of standardized global agriculture metrics for quantifying $CO_2$ emissions, which includes abatement across diverse CLS-A practices and basic economic trade of modern farm-level implementation in individual sectors of Turkey. Furthermore, it should focus on individual indicators of agricultural products in different regions of Turkey.

## Supporting information

**S1 File. Appendix.**
(DOCX)

**S2 Data. Datasheet.**
(XLSX)

## Author contributions

**Conceptualization:** Nawaz Khan.

**Data curation:** Nawaz Khan, Wang Jie.

**Funding acquisition:** Wang Jie.

**Investigation:** Wang Jie.

**Methodology:** Nawaz Khan.

**Resources:** Nawaz Khan, Wang Jie.

**Software:** Nawaz Khan, Wang Jie.

**Supervision:** Nawaz Khan, Wang Jie.

**Validation:** Nawaz Khan, Wang Jie.

**Visualization:** Nawaz Khan.

**Writing – original draft:** Nawaz Khan.

**Writing – review & editing:** Nawaz Khan.

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
