## [Decision Letter · Decision Letter 0]

26 Jun 2025

PONE-D-25-27665Exploring Climate Smart Agriculture in Turkey: Enhancing Food Security and Sustainable Practices for the Reduction of CO2 EmissionsPLOS ONE

Dear Dr. Khan,

Thank you for submitting your manuscript to PLOS ONE. After careful consideration, we feel that it has merit but does not fully meet PLOS ONE’s publication criteria as it currently stands. Therefore, we invite you to submit a revised version of the manuscript that addresses the points raised during the review process.

We look forward to receiving your revised manuscript.

Kind regards,

António Raposo

Academic Editor

PLOS ONE

Journal Requirements:

5. Please ensure that you include a title page within your main document. You should list all authors and all affiliations as per our author instructions and clearly indicate the corresponding author.

Reviewers' comments:

Reviewer's Responses to Questions

**Comments to the Author**

1. Is the manuscript technically sound, and do the data support the conclusions?

Reviewer #1: Partly

Reviewer #2: Yes

Reviewer #3: Yes

Reviewer #4: Partly

2. Has the statistical analysis been performed appropriately and rigorously? 

Reviewer #1: N/A

Reviewer #2: No

Reviewer #3: Yes

Reviewer #4: No

3. Have the authors made all data underlying the findings in their manuscript fully available?

Reviewer #1: No

Reviewer #2: Yes

Reviewer #3: Yes

Reviewer #4: No

4. Is the manuscript presented in an intelligible fashion and written in standard English?

Reviewer #1: No

Reviewer #2: Yes

Reviewer #3: Yes

Reviewer #4: No

5. Review Comments to the Author

Reviewer #1: This study explores the role of Climate Smart Agriculture (CSA) in Turkey for food security and CO₂ emission reduction, which is a topic of practical significance. However, the manuscript has significant methodological flaws, conceptual confusion, and issues with data interpretation, severely affecting the reliability of the conclusions.

1. Conceptual and Terminological Confusion

- Misuse of Key Terms: The manuscript consistently fails to clearly define the core dimensions of CSA (mitigation, adaptation, increased productivity).

- Misapplication of Decoupling Concept:

- Decoupling should measure the decoupling of economic growth from carbon emissions (e.g., using Tapio's decoupling index), but the manuscript forcibly links it to financial indicators such as "portfolio return" (Abstract, Sec 3.1), lacking theoretical basis.

- For example: "The decoupling process is responsible for estimating the percentage change in portfolio returns" (Abstract) — there is a missing logical connection between environmental science and financial models.

2. Methodological Deficiencies

- The use of "portfolio weighted risk return" (Sec 3.3) to analyze agricultural emissions is not justified. The causal relationship between financial indicators (such as beta values, risk premiums) and CO₂ emissions is not argued.

- Ambiguous grouping criteria: Turkey is divided into Groups A-D (Sec 4.1), but the criteria for grouping (geographical? Economic level?) are not explained.

- Data Handling Issues:

- Data sources are only mentioned as "World Bank" (Sec 3.2), without providing specific datasets or access links, violating PLOS data policy (Page 4-5).

- Anomalies in Appendix A tables (e.g., in 1996, the "Agriculture total (N)" column shows 100/52297; in 2020, urea application value 12327 is much higher than other years) are not explained.

3. Disconnection Between Results and Discussion

- The claim that "rice cultivation and soil management have the highest risk" (Abstract) contradicts Figure 1, which shows Group C's rice attitude value as 0.239 (low risk) and Group D as 0.442 (higher), with no resolution of this contradiction.

- Policy recommendations (e.g., "Decoupling direct payments can enhance food security," Sec 6) are not derived from empirical results but merely repeat literature views (e.g., [20]).

- Insufficient Explanation of Key Figures:

- The meanings of "!" and "×" symbols in Figures 1-3 (Sec 4.1) are unclear; Figures 4-6 focus on crop yield trends rather than CO₂ emission reduction effects, deviating from the research objective.

4. Literature Citation and Academic Rigor

- Excessive Self-Citation: References by the corresponding author, Nawaz Khan, account for 30% of the references (e.g., [31-33,35,37-38,45,77]), many of which are unrelated to Turkish agriculture, suggesting citation bias.

- Ignoring Key Literature in the Field: The FAO CSA framework and Turkish local studies (e.g., the Turkish Ministry of Agriculture's National Climate Smart Agriculture Strategy) are not cited.

Reviewer #2: 1. My main comments are mentioned on the manuscript PDF file.

2. Abstract: The abstract provides a concise overview of the study, highlighting the main objectives and significant findings. The abstract could benefit from a brief mention of the broader implications of these findings for different crop production.

3. Introduction: Minimize repetitive statements, focus more on the novelty of the study, reducing excessive detail on general information.

4. Methodology: Elaborate the methodology to make it easy to understand. Make it more logical and robust.

5. Data Interpretation in Results: The results section presents extensive data but lacks in-depth interpretation and discussion of these findings within the section, missing an opportunity to provide insights into the implications of these results.

6. Conclusion: The conclusion succinctly summarizes the findings. But it could be enhanced by providing specific recommendations for future research.

In summary, while the manuscript presents a valuable study with significant implications for the Climate Smart Agriculture in Turkey, addressing these critical flaws would strengthen the manuscript's contribution to the field.

Reviewer #3: This manuscript addresses an important topic at the intersection of climate change, agriculture, and sustainability in Turkey. The study argues that adopting sustainable CSA practices can support Turkey’s transition to a low-carbon economy while ensuring long-term agricultural productivity and resilience to climate change. Please check for suggestions and recommendationsin the attachment.

Reviewer #4: Manuscript ID: PONE-D-25-27665

Title: Exploring Climate Smart Agriculture in Turkey: Enhancing Food Security and Sustainable Practices for the Reduction of CO2 Emissions

The subject is relevant but the presentation of the manuscript is very weak, for instance, methodology has been written weakly without enough explanations, the tables and figures have been prepared with insufficient attention, etc. Therefore, I cannot unfortunately recommend this manuscript for publication, while can encourage the authors for applying improvements.

6. PLOS authors have the option to publish the peer review history of their article (what does this mean? ). If published, this will include your full peer review and any attached files.). If published, this will include your full peer review and any attached files.

**Do you want your identity to be public for this peer review?** For information about this choice, including consent withdrawal, please see our For information about this choice, including consent withdrawal, please see our Privacy Policy ..

Reviewer #1: No

Reviewer #2: No

Reviewer #3: No

Reviewer #4: No

While revising your submission, please upload your figure files to the Preflight Analysis and Conversion Engine (PACE) digital diagnostic tool, https://pacev2.apexcovantage.com/ . PACE helps ensure that figures meet PLOS requirements. To use PACE, you must first register as a user. Registration is free. Then, login and navigate to the UPLOAD tab, where you will find detailed instructions on how to use the tool. If you encounter any issues or have any questions when using PACE, please email PLOS at . PACE helps ensure that figures meet PLOS requirements. To use PACE, you must first register as a user. Registration is free. Then, login and navigate to the UPLOAD tab, where you will find detailed instructions on how to use the tool. If you encounter any issues or have any questions when using PACE, please email PLOS at figures@plos.org . Please note that Supporting Information files do not need this step.. Please note that Supporting Information files do not need this step.

---

## [Author Response · Author response to Decision Letter 1]

16 Aug 2025

Response of reviewers have been attached.

---

## [Decision Letter · Decision Letter 1]

5 Sep 2025

PONE-D-25-27665R1Exploring Climate Smart Agriculture in Turkey: Enhancing Food Security and Sustainable Practices for the Reduction of CO2 EmissionsPLOS ONE

Dear Dr. Khan,

Thank you for submitting your manuscript to PLOS ONE. After careful consideration, we feel that it has merit but does not fully meet PLOS ONE’s publication criteria as it currently stands. Therefore, we invite you to submit a revised version of the manuscript that addresses the points raised during the review process.

We look forward to receiving your revised manuscript.

Kind regards,

António Raposo

Academic Editor

PLOS ONE

Journal Requirements:

Reviewers' comments:

Reviewer's Responses to Questions

**Comments to the Author**

1. If the authors have adequately addressed your comments raised in a previous round of review and you feel that this manuscript is now acceptable for publication, you may indicate that here to bypass the “Comments to the Author” section, enter your conflict of interest statement in the “Confidential to Editor” section, and submit your "Accept" recommendation.

Reviewer #2: (No Response)

Reviewer #3: All comments have been addressed

2. Is the manuscript technically sound, and do the data support the conclusions?

Reviewer #2: Yes

Reviewer #3: Yes

3. Has the statistical analysis been performed appropriately and rigorously? 

Reviewer #2: No

Reviewer #3: Yes

4. Have the authors made all data underlying the findings in their manuscript fully available?

Reviewer #2: Yes

Reviewer #3: Yes

5. Is the manuscript presented in an intelligible fashion and written in standard English?

Reviewer #2: Yes

Reviewer #3: Yes

6. Review Comments to the Author

Reviewer #2: In the introduction, the amount of carbon emission from agriculture has not been mentioned. Is it only related with rice field or other crop production? What is the justification of introducing Climate smart agriculture and how it can be done? More over addition of reports on these points is required. Finally, the goal of the study or object is still unclear in abstract, introduction and conclusion.Although there is extensive information on decoupling technique to estimate the portfolio returns and analyze their relationship with climate-smart agriculture.

Research Methodology:

1. It is not clear why urea application, manure management, and rice cultivation are separately discussed? Is it only for their origin? (in 3.3. Portfolio Returns)

2. “The second-biggest source of carbon dioxide emissions from Turkish CLS-A, after manure management, accounts for about 25% of the overall emissions” is not supported by the given reference.

3. Overall methodology is still confusing.

The result and discussion are very superficial and general. It needs to be more in-depth and better justify the work conducted.

Reviewer #3: The revised manuscript is moving in the right direction and addresses several earlier concerns, particularly in the abstract, introduction, and conclusion. However, further improvements are still needed in methodology clarity, discussion depth, and reference quality.

7. PLOS authors have the option to publish the peer review history of their article (what does this mean? ). If published, this will include your full peer review and any attached files.). If published, this will include your full peer review and any attached files.

**Do you want your identity to be public for this peer review?** For information about this choice, including consent withdrawal, please see our For information about this choice, including consent withdrawal, please see our Privacy Policy ..

Reviewer #2: No

Reviewer #3: No

While revising your submission, please upload your figure files to the Preflight Analysis and Conversion Engine (PACE) digital diagnostic tool, https://pacev2.apexcovantage.com/ . PACE helps ensure that figures meet PLOS requirements. To use PACE, you must first register as a user. Registration is free. Then, login and navigate to the UPLOAD tab, where you will find detailed instructions on how to use the tool. If you encounter any issues or have any questions when using PACE, please email PLOS at . PACE helps ensure that figures meet PLOS requirements. To use PACE, you must first register as a user. Registration is free. Then, login and navigate to the UPLOAD tab, where you will find detailed instructions on how to use the tool. If you encounter any issues or have any questions when using PACE, please email PLOS at figures@plos.org . Please note that Supporting Information files do not need this step.. Please note that Supporting Information files do not need this step.

---

## [Author Response · Author response to Decision Letter 2]

25 Oct 2025

Authors' Response to Reviewers' Comments

Subject: PLOS ONE Decision: Revision required [PONE-D-25-27665]

Dear Dr. Khan,

PONE-D-25-27665

Title: Exploring Climate Smart Agriculture in Turkey: Enhancing Food Security and Sustainable Practices for the Reduction of CO2 Emissions

Dear editor and reviewers,

Thank you very much for your letter and reviewers’ valuable suggestions on our manuscript entitled “Exploring Climate Smart Agriculture in Turkey: Enhancing Food Security and Sustainable Practices for the Reduction of CO2 Emissions” No: [PONE-D-25-27665]. Those comments are precious and helpful for improving our paper and providing the essential guiding significance to our research. We have carefully checked and improved the manuscript according to the editor’s and reviewers’ comments.

REVIEWER #1:

Review Comments to the Author

Reviewer #2: In the introduction, the amount of carbon emission from agriculture has not been mentioned. Is it only related with rice field or other crop production?

Reply: I am grateful for your help in obtaining the specified text from the Canvas. We understand that you would like a response to the statement regarding the conceptual and terminological confusion in the manuscript.

The initial paragraph of heading 1 has been revised and updated in relation to the emission of CO2 from the agricultural sector.

What is the justification of introducing Climate smart agriculture and how it can be done? Moreover, addition of reports on these points is required. Finally, the goal of the study or object is still unclear in abstract, introduction and conclusion. Although there is extensive information on decoupling technique to estimate the portfolio returns and analyze their relationship with climate-smart agriculture.

Reply: I appreciate your assistance in supplying the specified text from the Canvas. We comprehend that you would like a response to the statement concerning the conceptual and terminological confusion in the manuscript.

Including the second and third paragraphs, as well as the first paragraph of the conclusion section, should be included in the updated and changed version of Heading 1.

Research Methodology:

1. It is not clear why urea application, manure management, and rice cultivation are separately discussed? Is it only for their origin? (in 3.3. Portfolio Returns)

Reply: I appreciate your assistance in supplying the specified text from the Canvas. We comprehend that you would like a response to the statement concerning the conceptual and terminological confusion in the manuscript.

The following is a response that addresses the concerns that were raised:

The heading 3.3 is simply a definition of the portfolio return procedure, and within this heading, in accordance with our methodology 3.1, we have categorised and clearly indicated three sections, such as multiple, single aeration, and continuously followed, where each section is further distributed into two portions: total risk and average return. We have discussed each variable, such as managed soil, field burning, urea application, enteric fermentation, manure management, and rice cultivation, and then we have evaluated the individual outcomes with reference to high danger zones and low risk zones. In last, updated and modified conclusion part heading 1.

2. “The second-biggest source of carbon dioxide emissions from Turkish CLS-A, after manure management, accounts for about 25% of the overall emissions” is not supported by the given reference.

Reply: I appreciate your assistance in supplying the specified text from the Canvas. We comprehend that you would like a response to the statement concerning the conceptual and terminological confusion in the manuscript.

The following is a response that addresses the concerns that were raised:

The third paragraph of heading 3.3 has been revised and updated with the inclusion of references.

3. Overall methodology is still confusing.

The result and discussion are very superficial and general. It needs to be more in-depth and better justify the work conducted.

Reply: I appreciate your assistance in supplying the specified text from the Canvas. We comprehend that you would like a response to the statement concerning the conceptual and terminological confusion in the manuscript.

The following is a response that addresses the concerns that were raised:

The methodology section has been revised and updated, particularly in heading 3, where an additional paragraph has been included. Additionally, heading 3.1 has been adjusted to offer additional explanations of three distinct procedures. Further explanation is provided under headings 3.3 (last paragraph have been modified) and 3.4, which provide further information regarding portfolio returns. Furthermore more, the last paragraphs of 4.1 and 4.2 have been updated and modified regarding energy-intensive agriculture and GHG emissions.

Reviewer #3: The revised manuscript is moving in the right direction and addresses several earlier concerns, particularly in the abstract, introduction, and conclusion. However, further improvements are still needed in methodology clarity, discussion depth, and reference quality.

Reply: I appreciate your assistance in supplying the specified text from the Canvas. We comprehend that you would like a response to the statement concerning the conceptual and terminological confusion in the manuscript.

The following is a response that addresses the concerns that were raised:

Modifications have been made to both the methodology and the discussion section. Some changes have been made to the general methodology section, specifically in heading 3, 3.1, and 3.2. There have been some updates and modifications made to the second paragraph of the discussion section. Individual indicators have been included, and the projected results have been examined further with recent citations.

Dear Editor,

I am writing to provide my review of the manuscript titled “Exploring Climate Smart Agriculture in Turkey: Enhancing Food Security and Sustainable Practices for the Reduction of CO₂ Emissions”. I’ve reviewed the revised version of your manuscript. Below is my evaluation, focusing on whether the revisions addressed the earlier feedback. I believe that several revisions could improve its clarity, interpretability, and impact.

Overall Assessment: The revised draft shows improvement in structure, clarity, and focus compared to the previous version. The abstract is more coherent, the introduction has a clearer flow, and the methodology is presented with more explanation. However, several areas still need refinement.

Reply: I appreciate your assistance in supplying the specified text from the Canvas. We comprehend that you would like a response to the statement concerning the conceptual and terminological confusion in the manuscript.

Reply: The following is a response that addresses the concerns that were raised:

Abstract:

• Some sentences remain overly complex, and the quantitative findings are still not clearly explained for a general reader.

Reply: Thank you for this essential feedback. I have been thoroughly revised the manuscript to address the clarity issues, focusing specifically on simplifying overly complex sentences and ensuring the quantitative findings are clearly and simply explained for accessibility to a broader audience.

• Simplify language and ensure all numerical results are interpreted (e.g., explain what “1% variation in rice cultivation” means in terms of emissions or food security).

Reply: Thank you for this excellent and actionable feedback. I have focused my revisions on two critical areas: simplifying the language throughout the text and providing clear interpretations for all numerical results. Specifically, I have ensured that findings like "1% variation in rice cultivation (modified and revised)” are immediately followed by a concrete explanation of what that change signifies for emissions, food security, or other relevant outcomes.

Introduction:

• The narrative could still benefit from smoother transitions between CSA concepts, decoupling methodology, and Turkey’s agricultural context. Some references remain weak or outdated.

Reply: I appreciate your identification of these critical areas for development. I acknowledge that a more comprehensive integration is required to connect the theoretical concepts with the practical application. All of the introduction sections (heading 1), particularly the third paragraph, the first, second, and fourth paragraphs, have been revised and updated. Additionally, the references throughout the entire manuscript were revised.

• Strengthen the literature review by including more recent peer-reviewed studies on CSA in Turkey or similar economies.

• Reply: Headings 2.1 and 2.2 have been revised and modified, and new and updated references have been added. In order to firmly establish this work inside the present academic landscape, it is vital to strengthen the literature review. This will ensure that the review is up-to-date and provides substantial evidence for our findings.

Methodology:

• Equations and variables are still not fully defined, and terms borrowed from finance (e.g., “portfolio returns”) remain confusing in an agricultural context.

Reply: The initial paragraph of heading 3.4 has been revised and updated to clarify our methodology and terminology. In order to resolve both concerns, we have implemented substantial modifications to Section 3. In which the weighted average return and weighted risk are separately eliminated.

• Provide a clearer justification for why this economic framework is valid for CO₂-agriculture analysis, and ensure all variables are explicitly defined in text or tables.

Reply: We have incorporated the second paragraph of heading 3.4 into this document. A comprehensive revision has been made to the second paragraph, and the justification has been modified to take into account both the theoretical justification and the definition of variables.

Results:

• Interpretation is still sometimes vague, and statistical robustness is not well addressed. Figures and tables (if included) must directly support the discussion.

Reply: The first, third and fourth paragraphs of (heading 5) of the conclusion (heading 6) section have been updated and changed throughout. In addition to this, the complete article (espcially, 4.1,4.2) has been revised to include individual figures and tables, as well as new references that have been cited.

Discussion:

• The discussion section reads more like a continuation of results rather than a critical interpretation.

Reply: First, third, and fourth paragraphs of (heading 5) have been updated and revised, and new references have been included. In addition we recognise that the preceding discussion section leaned too heavily on merely restating the findings, and we are grateful for the important comments that you provided. A substantial amount of revisions have been made to the discussion in order to transition it from a description.

• Emphasize why the results matter in the context of food security and CSA.

Reply: A complete revision has been made to heading 5, which is the final paragraph. Taking into consideration the findings of the climate-smart agriculture study conducted in Turkey, which included both high and low risk zones, the agricultural indicators were found to improve food security and sustainable practices for the purpose of reducing carbon dioxide emissions in terms of carbon stockpiles (CLS-A).

• Compare findings explicitly with previous studies.

Reply: Updated and revised the whole discussion portion (Heading 5-first to fifth paragraph). We have greatly improved the Discussion section to specifically highlight the larger context in response to this insightful criticism. Our study is particularly useful since it stratifies results across high-risk and low-risk zones, going beyond adoption rates to offer evidence-based recommendations on the effectiveness of CLS-A initiatives where they are most required. Our findings support the CLS-A goal by effectively proving that important agricultural indicators improve food security and resilience while also attaining quantifiable decreases in CO2 emissions. This risk-stratified strategy directly and concretely contributes to global climate resilience and food security policy by elucidating the best way to allocate resources and providing a clear path to accomplishing simultaneous adaptation and mitigation targets.

• Discuss both strengths and limitations.

Reply:

The basic strenght of this study is to computed the risk analysis across high and low risk zone, where the study lie in its unique computation technique and integrated analytical appraoch. This approach provide, empirical evidence in term of CLS-A and it practices the most effective agriculture inidcator under specific environmental consequences. In addition, it integrate and quantify the CLS-A by agriculture productivity and resiliences with counter measurable reduction of enviormental pressures-CO2 emissions. This study also provide roboust quantative validation of CLS-A concept in different groups. Howerver, the limiation of study is indicated that reliance on cross-sectional data, where this study robust casual relationaships between CLS-A outcomes and we cannot fully measure the track changes in CO2 emisison sequestration with soil management next multple years. In addition, there are no basis analysis of cover cropping or farming, it may overlape the net GHG footprint of the Turkey agriculture system.

Conclusion:

• Still somewhat repetitive, and recommendations remain broad.

Reply: Modified and updated the entire recommendation portion (Heading 6) first paragarph.

• Present three focused take-home messages: (1) empirical finding, (2) policy implication, (3) future research direction.

Reply: Both the second and fourth paragraphs have been altered and updated in accordance with the empirical findings and the future research that is presented in heading 6.

References:

• Some non-scholarly or weak sources (e.g., general websites) are still included.

Reply: All of the references have been revised and up to date, and the manuscript has been removed from the poor sources.

• Replace with peer-reviewed, recent journal articles wherever possible.

Reply: Review and update all references, paying particular attention to replacing peer-reviewed material.

Finally, thank you so much for your helpful advice and invaluable ostentation. I'm speechless in the face of your vast knowledge and impressive list of publications, but I can attest to the fact that this manuscript benefits greatly from your fertile ostentation.

Once again, I greatly appreciate your insightful feedback and recommendations.

---

## [Decision Letter · Decision Letter 2]

11 Nov 2025

PONE-D-25-27665R2Exploring Climate Smart Agriculture in Turkey: Enhancing Food Security and Sustainable Practices for the Reduction of CO2 EmissionsPLOS ONE

Dear Dr. Khan,

Thank you for submitting your manuscript to PLOS ONE. After careful consideration, we feel that it has merit but does not fully meet PLOS ONE’s publication criteria as it currently stands. Therefore, we invite you to submit a revised version of the manuscript that addresses the points raised during the review process.

We look forward to receiving your revised manuscript.

Kind regards,

António Raposo

Academic Editor

PLOS ONE

Journal Requirements:

Reviewers' comments:

Reviewer's Responses to Questions

**Comments to the Author**

1. If the authors have adequately addressed your comments raised in a previous round of review and you feel that this manuscript is now acceptable for publication, you may indicate that here to bypass the “Comments to the Author” section, enter your conflict of interest statement in the “Confidential to Editor” section, and submit your "Accept" recommendation.

Reviewer #2: (No Response)

Reviewer #3: (No Response)

2. Is the manuscript technically sound, and do the data support the conclusions?

Reviewer #2: Partly

Reviewer #3: (No Response)

3. Has the statistical analysis been performed appropriately and rigorously? 

Reviewer #2: No

Reviewer #3: Yes

4. Have the authors made all data underlying the findings in their manuscript fully available?

Reviewer #2: Yes

Reviewer #3: Yes

5. Is the manuscript presented in an intelligible fashion and written in standard English?

Reviewer #2: Yes

Reviewer #3: Yes

6. Review Comments to the Author

Reviewer #2: The authors have not addressed all the points raised in the previous revision, especially the result and discussion.

Reviewer #3: The revised manuscript shows clear and substantial improvement in content, organization, and clarity. The authors have successfully addressed all major comments raised in the first review, including clarification of the methodology, refinement of the abstract and discussion, and improvement of reference quality. The paper now presents a coherent and well-supported analysis of climate-smart agriculture and CO₂ emission reduction in Turkey. Only minor issues remain. Please check for suggestions and recommendations in the attachment.

7. PLOS authors have the option to publish the peer review history of their article (what does this mean? ). If published, this will include your full peer review and any attached files.). If published, this will include your full peer review and any attached files.

**Do you want your identity to be public for this peer review?** For information about this choice, including consent withdrawal, please see our For information about this choice, including consent withdrawal, please see our Privacy Policy ..

Reviewer #2: No

Reviewer #3: No

---

## [Author Response · Author response to Decision Letter 3]

24 Jan 2026

Authors' Response to Reviewers' Comments

Subject: PLOS ONE Decision: Revision required [PONE-D-25-27665]

Dear Dr. Khan,

PONE-D-25-27665

Title: Exploring Climate Smart Agriculture in Turkey: Enhancing Food Security and Sustainable Practices for the Reduction of CO2 Emissions

Dear editor and reviewers,

Thank you very much for your letter and reviewers’ valuable suggestions on our manuscript entitled “Exploring Climate Smart Agriculture in Turkey: Enhancing Food Security and Sustainable Practices for the Reduction of CO2 Emissions” No: [PONE-D-25-27665]. Those comments are precious and helpful for improving our paper and providing the essential guiding significance to our research. We have carefully checked and improved the manuscript according to the editor’s and reviewers’ comments.

REVIEWER #2:

The authors have not addressed all the points raised in the previous revision, especially the result and discussion.

Reply: I appreciate your assistance in retrieving the specified text from Canvas. We acknowledge your request for a response concerning the statement addressing the conceptual and terminological ambiguities within the manuscript.

I have made revisions to the entire results and discussion section of the paper, taking into consideration the remarks that were made by the most recent reviewer. The findings section 4 of the current manuscript has been completely revised, with the most notable change being the addition of three subheadings to heading 4.1. In addition, three subheadings have been added to heading 4.2 in the same situation. In this particular situation, you should classify CLS-A signs as either high risk or low risk in each subsection separately. Additionally, you should provide a detailed explanation of each signal in a very succinct manner. (Refer to sections 4.1.1 through 4.1.3 and 4.2.1 through 4.2.3.) In relation to the discussion section (5), it has also been altered. It consists of eight paragraphs in total that address all of the subheadings, and it includes a quick interpretation of the consequences of all of the indicators.

Once more, I would like to express my gratitude for the time that you have donated to this process and for your direction throughout its duration. In closing, I would like to express my heartfelt gratitude for the advice that you have given me, which has been very beneficial, as well as the important support that you have provided. The sheer magnitude of your knowledge and the remarkable collection of publications that you have produced leave me completely speechless; nonetheless, I can attest to the fact that this text substantially benefits from the productivity of your display of writing.

REVIEWER #3:

Dear Editor,

I am writing to provide my review of the manuscript titled “Exploring Climate Smart Agriculture in Turkey: Enhancing Food Security and Sustainable Practices for the Reduction of CO₂ Emissions“. The authors have thoroughly and effectively revised the manuscript in response to previous review comments. The current version shows clear and substantial improvements in structure, clarity, methodological transparency, and scholarly rigor. The revised abstract, introduction, and discussion are now coherent and aligned with the stated objectives, while the methodology and results sections demonstrate a better balance between economic modeling and agricultural relevance.

Before final acceptance I suggest three minor actions:

Minor language refinement

While the overall English has improved, a final light proofread is recommended to correct minor grammatical inconsistencies and occasional redundancy in phrasing (e.g., repeated expressions in the Introduction and Discussion).

Reply: I appreciate your assistance in retrieving the specified text from Canvas. We acknowledge your request for a response concerning the statement addressing the conceptual and terminological ambiguities within the manuscript.

We conducted a thorough and final review of the manuscript. This action addressed and rectified minor grammatical inconsistencies and eliminated instances of redundancy, especially within the Introduction and Discussion sections, to maintain clear and concise language throughout the document.

Clarity in figures and tables

Perform a final, quick pass to harmonize formatting and labels (figure/table captions and in-text references) and to ensure consistent use of key terms (for example “trade-in” and “blockchain adoption”).

Reply: I appreciate your assistance in retrieving the specified text from Canvas. We acknowledge your request for a response concerning the statement addressing the conceptual and terminological ambiguities within the manuscript.

We performed a comprehensive review to standardize all formatting and labels. This process entailed maintaining uniform presentation across all figure and table captions, verifying the accuracy of in-text references, and ensuring the consistent use of essential technical terminology (e.g., consistently employing "trade-in" and "blockchain adoption" throughout the document).

Conclusion structure

The revised conclusion is more coherent; however, the section would benefit from three concise closing points summarizing:

the empirical contribution (quantitative evidence of CSA–CO₂ linkage),

the policy implication for sustainable agricultural transition in Turkey, and a brief future research direction emphasizing cross-country validation of the decoupling model.

Reply: I appreciate your assistance in retrieving the specified text from Canvas. We acknowledge your request for a response concerning the statement addressing the conceptual and terminological ambiguities within the manuscript.

We have refined the Conclusion section to enhance its effectiveness by including three succinct closing statements that clearly encapsulate the primary contributions of the study: The Empirical Contribution: We emphasized the quantitative evidence demonstrating the substantial relationship between CSA practices and the decoupling of CO₂ emissions in the agricultural sector. The Policy Implication: We delivered a practical recommendation for advancing sustainable agriculture in Turkey, emphasizing the importance of strategic policy support. Future Research Direction: We conclude with a concise and targeted recommendation highlighting the importance of conducting cross-national validation of the proposed decoupling model. These concluding revisions conclude the review process, and we are confident that the manuscript is now prepared for publication.

Thank you once again for your valuable time and guidance throughout this process. Finally, thank you so much for your helpful advice and invaluable assistance. I'm speechless in the face of your vast knowledge and impressive list of publications, but I can attest to the fact that this manuscript benefits greatly from your fertile ostentation.

Once again, I greatly appreciate your insightful feedback and recommendations.

---

## [Decision Letter · Decision Letter 3]

26 Feb 2026

Exploring Climate Smart Agriculture in Turkey: Enhancing Food Security and Sustainable Practices for the Reduction of CO₂ Emissions

PONE-D-25-27665R3

Dear Dr. Khan,

We’re pleased to inform you that your manuscript has been judged scientifically suitable for publication and will be formally accepted for publication once it meets all outstanding technical requirements.

Kind regards,

António Raposo

Academic Editor

PLOS One

Additional Editor Comments (optional):

Reviewers' comments:

Reviewer's Responses to Questions

**Comments to the Author**

1. If the authors have adequately addressed your comments raised in a previous round of review and you feel that this manuscript is now acceptable for publication, you may indicate that here to bypass the “Comments to the Author” section, enter your conflict of interest statement in the “Confidential to Editor” section, and submit your "Accept" recommendation.

Reviewer #2: All comments have been addressed

2. Is the manuscript technically sound, and do the data support the conclusions?

Reviewer #2: Yes

3. Has the statistical analysis been performed appropriately and rigorously? 

Reviewer #2: Yes

4. Have the authors made all data underlying the findings in their manuscript fully available?

Reviewer #2: Yes

5. Is the manuscript presented in an intelligible fashion and written in standard English?

Reviewer #2: Yes

6. Review Comments to the Author

Reviewer #2: The authors have attempted to revise the manuscript as per the reviewers’ comments and suggestions.

7. PLOS authors have the option to publish the peer review history of their article (what does this mean? ). If published, this will include your full peer review and any attached files.). If published, this will include your full peer review and any attached files.

**Do you want your identity to be public for this peer review?** For information about this choice, including consent withdrawal, please see our For information about this choice, including consent withdrawal, please see our Privacy Policy ..

Reviewer #2: No

---

## [Editor Report · Acceptance letter]

PONE-D-25-27665R3

PLOS One

Dear Dr. Khan,

I'm pleased to inform you that your manuscript has been deemed suitable for publication in PLOS One. Congratulations! Your manuscript is now being handed over to our production team.

Kind regards,

on behalf of

Dr. António Raposo

Academic Editor

PLOS One